# Variables associated with owner perceptions of the health of their dog: Further analysis of data from a large international survey

**Richard Barrett-Jolley** [iD], **Alexander J. German** [iD]*

Institute of Life Course and Medical Sciences, University of Liverpool, Liverpool, United Kingdom

* ajgerman@liverpool.ac.uk

## Abstract

In a recent study (doi: 10.1371/journal.pone.0265662), associations were identified between owner-reported dog health status and diet, whereby those fed a vegan diet were perceived to be healthier. However, the study was limited because it did not consider possible confounding from variables not included in the analysis. The aim of the current study was to extend these earlier findings, using different modelling techniques and including multiple variables, to identify the most important predictors of owner perceptions of dog health. From the original dataset, two binary outcome variables were created: the *'any health problem'* distinguished dogs that owners perceived to be healthy ("no") from those perceived to have illness of any severity; the *'significant illness'* variable distinguished dogs that owners perceived to be either healthy or having mild illness ("no") from those perceived to have significant or serious illness ("yes"). Associations between these health outcomes and both owner-animal metadata and healthcare variables were assessed using logistic regression and machine learning predictive modelling using XGBoost. For the *any health problem* outcome, best-fit models for both logistic regression (area under curve [AUC] 0.842) and XGBoost (AUC 0.836) contained the variables dog age, veterinary visits and received medication, whilst owner age and breed size category also featured. For the *significant illness* outcome, received medication, veterinary visits, dog age and were again the most important predictors for both logistic regression (AUC 0.903) and XGBoost (AUC 0.887), whilst breed size category, education and owner age also featured in the latter. Any contribution from the dog vegan diet variable was negligible. The results of the current study extend the previous research using the same dataset and suggest that diet has limited impact on owner-perceived dog health status; instead, dog age, frequency of veterinary visits and receiving medication are most important.

## Introduction

There is an increasing interest from owners in feeding unconventional diets, such as those utilising uncooked (so-called 'raw' diets) or plant-based (either vegetarian or vegan diets) ingredients. In a recent UK survey, 7% and approaching 1% of pet owners reportedly fed their dog either a raw or vegan diet, respectively [1]. Veganism is an increasingly popular food choice in humans, with recent surveys indicating 2–3% [2], 4% [3] and 3% [4] of people in the UK,

**Data Availability Statement:** The original research data are accessible at https://osf.io/nbepu. All statistical reports and codes are available in the

Supporting Information and online (https://github.com/RichardBJ/CanineHealth23).

**Funding:** The authors received no specific funding for this work.

**Competing interests:** AJG is an employee of the University of Liverpool, but his position is financially-supported by Royal Canin. AJG has also received financial remuneration and gifts for providing educational material, speaking at conferences, and consultancy work, all unrelated to the current study. Royal Canin had no involvement in any aspect of the current work including study design, data analysis, drafting the manuscript or the decision to submit the work for publication. RBJ is an employee of the University of Liverpool, whose salary is financially-supported by the Higher Education Funding Council for England (HEFCE). RBJ also receives a stipend from United Kingdom Research and Innovation for work as a panellist for the Biotechnology and Biological Sciences Research Council (BBSRC). Neither HEFCE nor BBSRC had any involvement in any aspect of the current work including study design, data analysis, drafting the manuscript or the decision to submit the work for publication. There are no patents, products in development or marketed products associated with this research to declare. This does not alter our adherence to PLOS ONE policies on sharing data and materials.

Europe and USA, respectively, to be vegan. Although still uncommon (<1% of owners), use of plant-based (vegan or vegetarian) diets for dogs is increasing in popularity [1], not least amongst owners who are vegan themselves [5]. Interestingly, the popularity of feeding raw diets is also increasing, with the same survey suggesting that 7% of the UK dog-owning population use this method [1], a proportion similar to that reported in recent international study, where 9% of dog owners exclusively fed raw food [6]. The main reason that owners cite for considering a plant-based diet is concern about animal welfare [5], whilst owners who feed raw diets believe these to be more 'natural' and healthier than commercial food despite an absence of evidence to support this [7]. Despite their increasing popularity, concerns remain regarding the safety of feeding unconventional foods; for both vegan [5,8] and raw [9–11] food, in terms of nutritional adequacy. For raw food, there are additional concerns about contamination, both with pathogenic bacteria and bacteria resistant to multiple antimicrobials [11–13].

In recent studies, the safety of feeding unconventional diets for dogs has been examined, albeit using owner reports of health from questionnaires or surveys. In one study of 16,475 households that fed raw food, only 39 (0.2%) recalled a member of the household acquiring a pathogenic infection during the time they were using the food [14], perhaps, suggesting that risks to owners from bacterial contamination of raw food are uncommon. The health of dogs being fed unconventional diets has also been assessed. In one survey of 1,413 owners conducted across North America, those feeding plant-based diets reported fewer health conditions in their dogs and a longer lifespan, compared with owners of dogs fed a meat-based diet [6]. In a second study [15], owner-reported health of dogs being fed various diet types was assessed. Compared with dogs fed a conventional diet, owners that fed either raw food or a vegan diet reported better health. However, besides being reliant on owner-reported health and their recollections of other health-related information (e.g., medication usage; the presences of specific health disorders), rather than more objective health measures (e.g., verified diagnoses made by a veterinarian; information gathered from electronic patient records), no account was taken of possible confounding from other variables. To the authors' credit, the original study data were made available for use by other researchers (https://osf.io/nbepu), which not only included the diet and health data they analysed, but also other metadata including owner (e.g., country, career, education, age and gender) and animal (e.g., age, sex, neuter status and breed) variables, and healthcare-related variables (e.g., veterinary visits, receiving medication, use of therapeutic diet). Therefore, the aim of the current study was to extend the earlier findings using different statistical techniques to create models that best predicted owner perceptions of health, and to identify the relative importance of the variables that contributed to the final model. This included the use of binary logistic regression and machine learning predictive modelling using a scalable tree boosting system (XGBoost [16]).

## Materials and methods

### Study design

A secondary analysis was conducted using data from a recent international survey of owner opinions on the health of their dog [15]. The open-access dataset from this study is accessible at https://osf.io/nbepu. The reason for conducting this further analysis was that many variables gathered were not analysed statistically, and that the effect of possible confounding was not adequately investigated, both of which were limitations acknowledged by the study authors [15]. Further, we were particularly interested in exploring associations between a wider range of variables and owner perceptions of the health of their dog.

## Dataset

Full details of survey design, methodology, information collected and category definitions have been reported previously [15]. Briefly, the original survey was designed using an online survey platform (https://www.onlinesurveys.ac.uk), with owners being asked to record details about their dog (including demographic data, main diet type and health characteristics) and themselves. Owner information used for the current study included location (country or continental region), setting (urban, rural or a mix of urban and rural), education level (qualifications achieved), household income, whether their occupation was in an animal-related career (including veterinarian, veterinary technician or nurse, animal breeder, animal trainer or pet industry worker), age, gender and diet. Animal information used for the current study included: age, sex, neuter status, breed size category (e.g., toy, small, medium, large and giant) and main diet type (including conventional, raw, vegetarian, vegan, *in vitro* meat, insect, fungal or algal). We also used responses from owners about the healthcare their dog received including number of veterinary visits in the last 12 months, whether the dog had received medication (besides those used in preventive care e.g. routine vaccinations and endo- or ecto-parasite treatments) in the last 12 months, and whether the dog was on a therapeutic diet; when dogs were fed a therapeutic diet, owners were asked to base the information on their dog's diet on their previous diet [15]. Finally, in the original survey, owners were also asked their opinion about the health of their dog in the last 12 months, with possible responses including: 'healthy', 'generally healthy with minor or infrequent problems', 'significant or frequent problems', 'seriously ill' or 'unsure'. Since we were primarily interested in owner perceptions of health, we used this information to create our outcome variables, as described below.

## Statistical methods

**Statistical software.** All data pre-processing and statistical analyses were conducted using open-source statistical software (R, version 4+) [17], with several packages including: 'aod' version 1.3.2 [18], 'broom' version 0.8.0 [19], 'car' version 3.0.13 [20], 'caret' version 6.0.93 [21], 'cluster' version 2.1.4 [22], 'clusterSim' version 0.51.3 [23], 'corrplot' version 0.92 [24], 'data.table' version 1.14.2 [25], 'dbscan' version 1.1–11 [26], 'DescTools' version 0.99.45 [27], 'dplyr' version 1.0.9 [28], 'factoextra' version 1.0.7 [29], 'fmsb' version 0.7.3 [30], 'foreign' version 8.82 [31], 'ggforce' version 0.4.1 [32], 'ggplot2' version 3.3.6 [33], 'ggthemes' version 4.2.4 [34], 'glmtoolbox' version 0.1.3 [35], 'heatmaply' version 1.4.0 [36], 'Hmisc' version 4.7–0 [37], 'IHW' version 1.26.0 [38]. 'mltools' version 0.3.5 [39], 'pROC' version 1.18.0 [40], 'RColorBrewer' version 1.1–3 [41], 'readxl' version 1.4.0 [42], 'reshape2' version 0.8.9 [43], 'ROCR' version 1.0–11 [44], 'rsample' version 1.1.0 [45], 'rstatix' version 0.7.0 [46], 'showtext' version 0.9–5 [47]; 'smotefamily' version 1.3.1 [48], 'stringr' version 1.4.0 [49], 'tidyverse' version 1.3.1 [50], 'uwot' version 0.1.16 [51] and xgboost version 1.6.0.1 [16]. Statistical reports of all data pre-processing and analysis are included in the supplementary information (S1–S9 Files), whilst all code is available online (https://github.com/RichardBJ/CanineHealth23).

**Initial dataset pre-processing.** Data were imported into the statistical software using the 'readxl' package [42] and manipulated using the 'dplyr' package [28], with columns renamed for use in R (see below). Counts of all categorical data were examined, and categories with small groups were either removed or combined with others to exclude missing data or to reduce dataset imbalance, as described below. The variables included in the dataset were owner variables and animal variables (collectively owner-animal metadata), as well as healthcare variables and health outcomes variables.

Data pre-processing occurred in two stages: in the first stage (prior to Uniform Manifold Approximation and Projection; UMAP; see below), the original dog sex variable (male sexually

intact, male castrated, female sexually intact, female spayed) was separated into two binary variables, *dog sex* (male, female) and *neuter status* (yes, no). In addition, data were excluded from dogs where age was reported as 'unsure' (n = 3) or were aged <1 year (n = 26), to ensure a focus on adult dogs.

To facilitate the modelling studies (logistic regression and XGBoost), further data pre-processing steps were undertaken after UMAP. For owner variables, pre-processing included combining categories with small numbers in the *location* variable (e.g., 'Africa' [n = 7], 'Asia' [n = 25], 'other' [n = 11] and 'South America' [n = 47]) into a new 'other' category, and also removing the 'other' category (n = 15) from the *settings* variable. The *owner education* variable was simplified by combining the 'basic' (n = 15) and 'high school' (n = 478) categories and combining the 'PG' (n = 515) and 'PhD' (n = 90) categories. A binary *animal career* variable was created from the *career* variable by combining the 'veterinarian' (n = 125), 'veterinary technician or nurse' (n = 54), 'animal breeder' (n = 17), 'animal trainer' (n = 140) and 'pet industry worker' (n = 205) categories into a 'yes' category, whilst the remaining option ('none of the above' [n = 2,137]) formed the 'no' category. The *owner income* variable was simplified by removing the 'prefer not to answer' category (n = 184) and ordered, whilst the *owner age* variable was simplified (by combining the '18-19y' [n = 19] and '20-29y' [n = 391] categories into a new '<30y' category and combining the '60-69y' [n = 364] and '>70y' [n = 85] categories into a new '>60y' category) and then ordered. Given small numbers, the 'prefer not to answer' (n = 17) and 'other '(n = 4) categories were removed from *owner gender* variable, and the 'other' category (n = 18) was removed from the *owner diet* variable. Further, binary variables were created for owners who reported consuming a vegan diet (yes vs. no) and owners who reported consuming either a vegan or vegetarian diet (yes vs. no).

For animal variables, in addition to having a continuous *dog age* variable, an ordinal variable was also created with data grouped into quintiles (1-3y, 3-5y, 5-7y, 7-9y, 9-20y). Further, a *giant breed* binary variable (yes, no) was created from the *breed size category* variable. Moreover, a *dog diet* variable was created from owner responses to a question where owners selected from different diet categories based on a question where they were asked "to consider the main ingredients within your pet's normal diet ('meat' includes land animals, poultry and fish)" [15]. The 'unsure' (n = 8), 'insect-based' (n = 5), 'meat-based (lab-grown)' (n = 6) and 'mixture' (n = 13) categories were all removed since numbers were again small. This left four diet categories whose names were adapted from the original study [15] as follows: 'vegan' (original category: "vegan [consuming no animal products]"); 'vegetarian' (original category: "vegetarian [including eggs or milk, but not meat]"); 'raw' (original category: "meat-based raw"); and 'conventional' ("meat-based conventional"). From these same categories, binary variables were created for dogs consuming a *vegan* (yes vs. no), *vegan or vegetarian* (yes or no) or a *raw* (yes vs. no) diet.

For healthcare variables, the 'unsure' category (n = 13) was removed from the *veterinary visits* variable, and two ordinal variables were created; the first comprised 5 categories ('0 visits', '1 visit', '2 visits', '3 visits', '4+ visits'), whilst the second comprised 4 categories ('0 visits', '1 visit', '2 visits', '3+ visits' [n.b., '3 visits' and '4+ visits' categories combined]).

For our outcome variables (owner's perception of the health of their dog), we first removed data from owners who reported that they were 'unsure' (n = 5). From this, two binary health status variables were created: the first was an *any health problem* variable, where dogs classified as 'healthy' were classified as 'no', and all remaining categories ('generally healthy with minor or infrequent problems', 'significant or frequent problems' and 'seriously ill') were classified as 'yes'; the second was a *significant illness* variable, whereby dogs classified either as 'healthy' or 'generally healthy with minor or infrequent problems' were classified as 'no', and those classified as having 'significant or frequent problems' or being 'seriously ill' were classified as 'yes'.

As a final pre-processing step, a *decision-maker* (primary vs. other) variable was created from data about the role owners played choosing a diet for their dog; for this, the 'primary decision-maker' category was classified as 'primary', whilst the other two categories ('play no role', n = 15; 'play some lesser role', n = 96) were classified as 'other'. Given concerns about the reliability of information from owners who were not primary decision-makers and healthcare received, its effect on the outcome variables used in modelling studies (both binary illness variables) was first assessed by logistic regression, before the other modelling studies, as described below.

**Owner-dog metadata visualisation with uniform manifold approximation and projection with density-based spatial clustering.** Unsupervised uniform manifold approximation and projection (UMAP) [52] was used to reduce the multivariate owner and dog metadata data to two dimensions for visualisation purposes (using the 'uwot' package [51]). After this, density-based spatial clustering (DBSCAN, using the 'dbscan' package [26]) was applied to the UMAP outputs to identify potential groupings within these owner-animal metadata, prior to the data pre-processing necessary for the statistical and machine learning modelling, as explained above. The UMAP approach was specifically developed to be applicable to mixed datasets with categorical, ordinal and numeric data [52]. It is a potent, non-linear dimension-reduction technique that outperforms principal component analysis (PCA) and t-distributed stochastic neighbour embedding (TSNE) in its ability to preserve the global and local structure of data [52]. It employs a specific type of manifold projection known as Riemannian geometry, which has the strength of enabling meaningful clusters to be identified and visualised in a reduced dimensional space. This approach differs from PCA in that it typically reduces multivariate data to two parameters (arbitrarily called x and y); further, rather than calculating variability, UMAP instead non-linearly transforms the data to create a lower-dimensional representation, allowing more complex data patterns to be captured, compared with linear methods like PCA. The axes in UMAP do not represent 'principal components' and cannot be interpreted in the same way; instead, they help visualise high-dimensional data in a lower-dimensional space.

Each record (individual row of the dataset with one owner-dog combination) contributes one point to the UMAP graph. The following animal variables were included: *dog age*, *dog sex*, *neuter status*, *diet* and *breed size category*. In addition, the following owner variables were included: *owner age*, *owner gender*, *owner diet*, *location*, *setting*, *education*, *income* and *decision-maker status*. Following best practise [53], different distance metrics were used including Euclidian (for continuous data e.g., *dog age*), Manhattan (for ordered categorical data e.g., *owner age*, *education*) and Hamming (for non-ordered categorical data e.g. *dog diet*, *dog sex* etc).

The quality of DBSCAN clustering was assessed with a silhouette plot along with silhouette scores [53] and also by calculating Davies-Bouldin's cluster separation measure [54]. The silhouette plot provides a visual representation of how well points within the UMAP have been classified into their respective clusters, whilst average silhouette widths (silhouette scores) quantify how similar the point is to its own cluster relative to other clusters; silhouette scores can range from +1 to -1, with positive and negative values indicating good and poor matching, respectively; further, average cluster silhouette scores of 0.26–0.50, 0.51–0.70 and 0.71–1.00 represent weak, reasonable and strong clustering, respectively [55]. Davies-Bouldin's cluster separation measure is determined from the ratio between the scatter within and the separation between clusters, with smaller values suggesting better clustering and values <1 representing very good clustering.

**Visualising associations amongst owner-animal metadata and healthcare variables.** After the second data pre-processing stage, Kendall's *tau* [56] was used to conduct exploratory

correlation analysis using the R package 'rstatix' [46], and a correlation matrix [57] was plotted with the 'corrplot' package [24]. Data were first coerced into a numerical ordinal form, except for the *location* variable because there was no ordinal relationship amongst categories; instead, this variable was one-hot encoded, whereby a new binary variable was added for each category. *P* values are expressed as their negative logarithm (-Log*P*), whilst false discovery rate (FDR) was calculated by the Benjamini-Hochberg procedure using the 'IHW' package [38]. Hierarchical clustering was added with R stats::hclust using the Ward method [58]. Strengths of association were reported according to Cohen [59], whereby correlation coefficients of 0.1–0.3, 0.3–0.5 and 0.5–1.0 represented small, medium and large correlation, respectively.

**Binary logistic regression.** Binary logistic regression models were created using the 'glm' function in R. Both binary illness variables (*any health problem* and *significant illness*) were used separately as outcome variables, whilst all owner, animal and healthcare variables were tested as predictor variables. We used a combination of simple and multiple logistic regression. Simple logistic regression analyses were initially performed using data from all owners (irrespective of decision-maker status), with results shown in the supplementary information (S1 and S2 Tables). However, these models revealed concerns over the reliability of data from owners who were not primary decision-makers (see below) and, as a result, all remaining logistic regression analyses were instead conducted with the primary decision-maker dataset. For these analyses, the dataset was first randomly divided into training (75%) and test (25%) datasets, using the 'sample' function in R, a step that was necessary to facilitate model validation by receiver operating characteristic (ROC) curves (see below).

Simple logistic regression models created with the primary decision-maker dataset enabled unadjusted associations between each outcome variable and single predictor variables to be determined. Some variables had been coded in different ways, as described above; for example, *dog age* was coded as both a continuous variable and an ordinal variable with 5 categories, with the continuous *dog age* variable also being tested with and without basis (b-)splines; further, veterinary visits were coded as both a 4- or 5-category variable, whilst *dog diet* was coded both as a 4-category variable [conventional, raw, vegetarian and vegan] or as separate binary variables for *raw diet*, *vegan diet* or a combined *vegan-vegetarian* binary variable [see above]. To determine the coding approach that was most appropriate for such variables, performance of competing models was assessed using the Bayesian Information Criterion (BIC, see below) [60]; the coding approach that fitted the data best was selected for use in the initial multiple regression modelling as long as model assumptions were met (e.g., independence of errors, the requirement for a linear relationship between the logit of the outcome and continuous predictor variables). For example, the *vegan diet* variable performed better than the *vegan-vegetarian diet* variable, the continuous *dog age* variable performed better than the ordinal *dog age* variable, albeit that b-splines (utilising boundary knots and an internal knot at the median value [6 years]) were required to ensure that model assumptions were met. Further, for the *significant illness* outcome variable, the *giant breed* predictor variable performed better than the *breed size category* variable, and there were also some issues with model convergence for the latter. Models containing two related predictor variables, with and without interaction terms, were also tested when these were clinically relevant (e.g., between dog sex and neuter status, between owner and dog diet categories and between location and setting).

In the first step of the multiple logistic regression modelling, two multiple regression models were created for each outcome variable (*any health problem* and *significant illness*); all predictor variables were included in the first of these models (*all-variable* model), whilst only owner and animal variables were included in the second model (*owner-animal metadata* model). The performance of *all-variable* and *owner-animal metadata* models were then compared with each other and with the *best-fit* models that were created subsequently. For these

*best-fit* models, we used a supervised, stepwise approach starting with all variables. This initial model was manually refined in both a backwards and forwards stepwise fashion, with the BIC (a measure of its goodness of fit compared with its complexity) being used to select the model within the same family with the best generalisability [60]. Using this approach, variables could be added or removed until the model with the smallest BIC was found, according to previously published rules [61], whereby model generalisability was deemed to be superior if the BIC of the new model was less than the previous model by at least 2 units.

After selecting a *best-fit* model and, given the importance of diet in the original work [15], both the *dog diet vegan* and *owner diet vegan* variables were separately added back, to determine their effect on model performance.

Results are reported as estimates of the regression coefficients (β) with the associated standard error (SE), and with odds ratios (OR) with the associated 99% confidence intervals (99%-CI). Influential datapoints were identified and assessed using Cook's distance and, in most models, none were identified. In the occasional cases where outliers were identified, we checked them in the original data to make sure there were no obvious errors (e.g., unrealistic results such as dog age >20 years); no such errors were found. Given that we were using data from a secondary source, and had no further means of verifying them (e.g., by cross-checking against the original questionnaire response), we decided not to remove them as there was no valid justification to do so (e.g., a typographical error created when the original data were entered). Possible multicollinearity in all models was assessed using the generalised variance inflation factor (GVIF) and GVIF(1/(2×*Df*); these were deemed to be acceptable when all values were <4 and <2 for GVIF and GVIF(1/(2×*Df*), respectively [62]. If necessary, multicollinearity was resolved by removing the variable with the greatest GVIF. Goodness of fit was tested by a visual inspection of observed and expected results and the Hosmer and Lemeshow test for large datasets. In addition, the proportion of variance explained in the overall model was assessed by the coefficient of determination (pseudo-$R^2$) based on the method reported by Nagelkerke [63]. Further, the log-odds of the outcome variable and dog age was plotted to enable visual assessment of whether the relationship was linear, and this was further assessed with the Box-Tidwell test. Finally, prediction accuracy was assessed by creating ROC curves, using the ROCR package [44], and then calculating their area under the curve (AUC). Values could range from 0 to 1; a model that performed no better than chance would have an AUC of 0.5, and models predicting better than by chance would have AUC >0.5, with an AUC of 1.0 suggesting perfect prediction. The test dataset was used to generate the ROC curves and AUC for models using data from owners who were primary decision-makers; however, because the *all-owner* dataset was not subdivided, AUC are based in ROC curves from the original dataset.

**Machine learning predictive modelling.** A scalable tree boosting system (XGBoost; "eXtreme Gradient Boosted" tree) was implemented in the statistical environment [16]. Data were pre-processed as described above and, to facilitate XGBoost, ranked variables used for the Kendall's tau correlation (as described above) were coerced to a numeric datatype, using one-hot encoding where necessary, and then subdivided 70:30 into training and test sets. A model was then built on the training data without any involvement of the test set. After pre-processing but before training, training data were further divided into training and validation subsets (70:30), and the training subset augmented as per standard practice for machine learning [64]. We trialled two augmentation methods: SMOTE [48] and a similar method where underrepresented classes were duplicated, but with simple addition of signal noise. Final models used the latter given that it performed better than SMOTE. Test data were left untouched. The model was trained using the augmented training data, and then tested against the validation training subset with area-under the curve (AUC) as the evaluation metric, enabling the final model to be built. Receiver operating characteristic curves were calculated and plotted

with the saved model against the previously unseen test dataset using the packages 'pROC' [40] and 'ROCR' [44]. Prediction accuracy was estimated by calculating the ROC AUC in a similar manner to that described above for logistic regression. Different measures of the contribution of the variables ('features') included in the final models were also calculated and depicted graphically. These included 'feature importance' (a metric of the fractional contribution of each feature to the model, based on the total gain from including each feature), 'cover' (a metric of the number of observations related to this feature in the model) and 'frequency' (a metric that represents the relative number of times a feature has been used in trees). Since these metrics are normalised, the sum of scores for all features in the model is 1.0.

## Results

### Owner-dog metadata visualised by uniform manifold approximation and projection

The original dataset comprised information on 2,639 owner-dog pairs [15]. After the initial pre-processing stage to create separate dog sex and neuter status variables, and also removing dogs without age data (n = 3) or that were <1 year of age (n = 27), the number of owner-dog pairs with available data was 2,609. We first used a UMAP projection and density-based clustering to create a simple overall visualisation of relationships in these owner-animal metadata (Fig 1; S1 and S2 Figs). Six distinct clusters were identified, with a Davies-Bouldin's cluster separation measure of 0.652 (good). Good clustering was evident within 3 clusters (average silhouette scores: cluster 2, 0.86; cluster 5, 0.84; cluster 6, 0.80), there was reasonable clustering in two others (average silhouette scores; cluster 3, 0.62; cluster 4, 0.59), but clustering was poor in the final cluster (cluster 1, average silhouette score 0.36).

Examining the owner-animal metadata, cluster 2 was most distinctive, almost-exclusively comprising dogs fed either a vegan (328/374 [87.7%] points in the cluster, 97.6% of all vegan dogs) or vegetarian (31/374 [8.3%] points in the cluster, 88.6% of all vegetarian dogs) diet, and also containing a predominance of owners consuming a vegan diet (332/374 [88.8%] points in the cluster, 57.1% of all vegan owners). In contrast, cluster 3 comprised owners who consumed either a vegan (249/608, 41.0%), vegetarian (240/608, 39.5%) or pescatarian (119/608, 19.6%) diet, but fed their dog either a conventional meat-based (423/608, 69.6%) or raw (175/608, 28.8%) diet. Cluster 6 was also distinctive in that it mostly comprised owners who consumed an omnivore (214/316, 67.7%) or omnivore reducing (98/316, 31.0%) diet and fed their dogs a raw diet (299/316, 94.6%). In the remaining 3 clusters (clusters 1, 4 and 5), most owners consumed either an omnivore or omnivore-reducing diet and usually fed their dog either a conventional meat-based diet or raw diet. Besides owner and dog diet, there were no major differences amongst clusters for the other variables, although some variables did appear to separate some clusters (e.g., cluster 4 for the setting variable; cluster 5 for the location variable; S1 and S2 Figs).

### Summary of owner-animal metadata

After the further data pre-processing stage, as described above, the number of owner-dog pairs with available data was 2,322, and details are shown in Table 1. This population was similar to the data used in the previous study [15]. Briefly, owners represented a broad age range with a relatively even distribution in education level, from high school to postgraduate. However, most (2,146, 92.4%) respondents were female, whilst most responses were from the United Kingdom (1,659, 71.4%), with other European countries the next most common region represented. Almost a quarter of the respondents (532, 22.9%) defined themselves as vegan with a

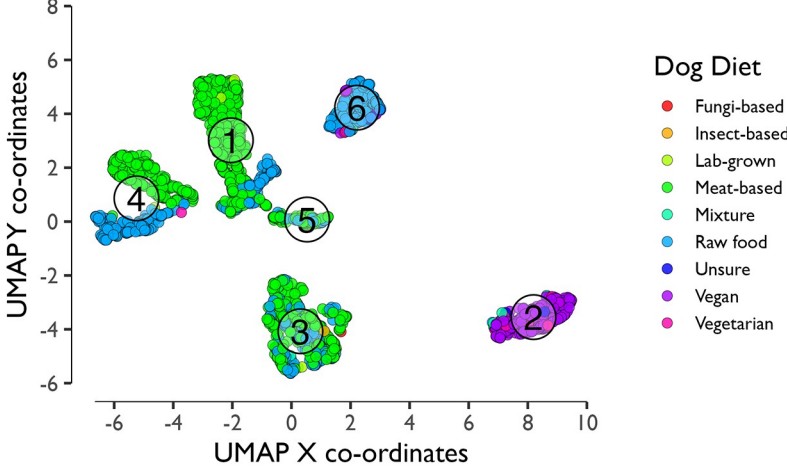

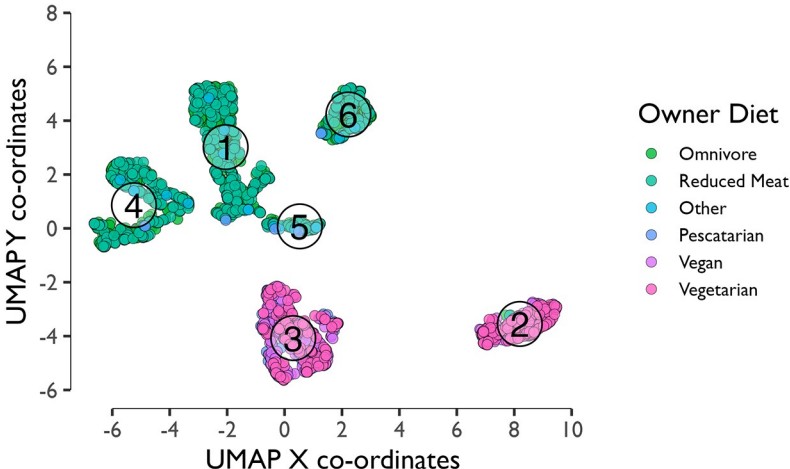

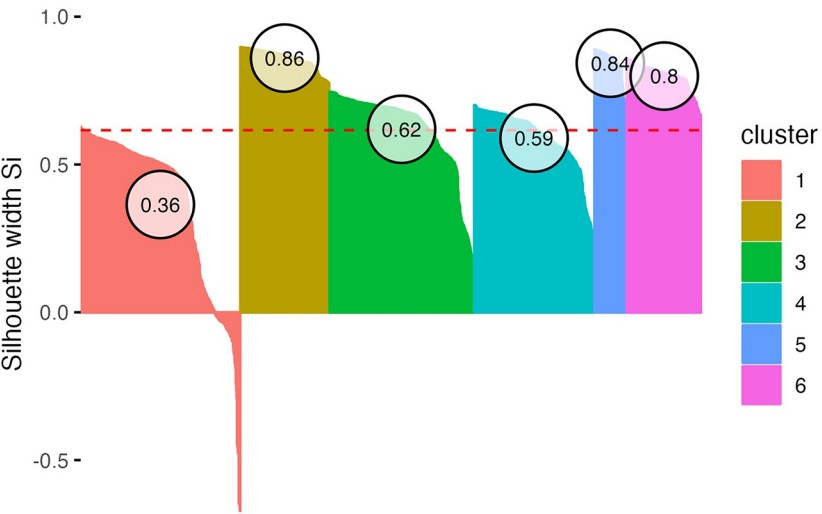

**Fig 1. Visualisation of owner metadata by uniform manifold approximation and projection with density-based spatial clustering.** Metadata for 2,609 owner-dog pairs were pre-processed with all factors as numeric and subject to dimension reduction with the UMAP projection technique. Owner variables included in this visualisation were *diet*, *sex*, *location*, *education* and *income*; animal variables included were *age*, *sex*, *neuter status*, *diet* and *breed size category*. Healthcare variables were not included. Each individual row of the data (owner-dog combination) contributes one UMAP x and y coordinate. Points colour-coded by *dog diet* (a) or *owner diet* (b), where 5 distinct clusters are evident. (c) Silhouette plot depicting how well clusters were separated within the UMAP; silhouette values measure how similar the point is to its own cluster relative to other clusters, with positive and negative values indicating good and poor matching, respectively. The numbers in circles are the average silhouette score for each cluster, whereby values of 0.26–0.50, 0.51–0.70 and 0.71–1.00 representing weak, reasonable and strong clustering, respectively.

further 235 (10.1%) reporting to be vegetarian. Regarding the dogs in this dataset (Table 1), a range of ages and breed sizes were included with the majority of male (294/1169, 74.9%) and female (192/1042, 81.6%) dogs being neutered. Approximately a third (744, 32.0%) of dogs were fed a raw meat diet, whilst 313 (13.5%) and were 35 (1.5%) fed vegan or vegetarian diets, respectively. Finally, most owners (2,211; 95.2%) reported that they were the primary decision-makers for their dog's diet (Table 1).

We were concerned with the reliability of data from the 111 owners (4.8%) who were not the primary decision-makers, because their knowledge about the health of their dog could well be less reliable and their beliefs and behaviours might also be different, all of which would likely impact perceptions about the health of their dog. To examine this, we created separate binary logistic regression models for the *any health problem* and *significant illness* outcome variables, with *decision-maker status* as the sole predictor variable. There was no difference in the odds of an owner recording their dog as having a *serious illness*, whether they were a primary decision-maker or not (OR 0.662: 99%-CI 0.286, 1.896; $P = 0.252$, pseudo-$R^2$ 0.0015). However, the odds of primary decision-makers recording their dog as having *any health problem* were less than for owners who were not primary decision-makers (OR 0.576: 99%-CI 0.347, 0.958; $P = 0.005$, pseudo-$R^2$ 0.0046). As a result of this difference, we decided to focus further analyses on data from the 2,211 owners who were primary decision-makers. However, to determine the effect of this decision, simple logistic regression was also performed on the data from all owners (irrespective of decision-maker status; S1 and S2 Tables), enabling comparisons to be made with simple logistic regression performed on the data from primary decision-makers only (see below).

## Summary of healthcare variables

Details of healthcare variables are also given in Table 1. Most dogs had visited their veterinarian at least once in the last year (all owner dataset 1,914 [82.4%]; primary decision-maker dataset 1,812 [81.9%]), whilst smaller proportions of dogs had received medication (all owner dataset 936 [40.3%]; primary decision-maker dataset 880 [39.8%]) and or had been switched to a therapeutic diet (all owner dataset 111 [4.8%]; primary decision-maker dataset 100 [4.5%]).

## Associations amongst owner-animal metadata and healthcare variables

Results of Kendall's tau correlations are shown in Fig 2, whilst datasets of all correlation coefficients and FDR-corrected -Log*P* are presented in the supplementary information (S3 and S4 Tables, respectively). Several significant associations were identified including a large positive association between owners and dogs consuming *vegan diets* (Kendall's tau 0.66, -Log*P* 212.0 [*P*<0.001]), a small positive association between UK location and dogs being fed a *raw diet* (Kendall's tau 0.24, -Log*P* 29.5 0 [*P*<0.001]), as well as positive associations amongst healthcare variables (*received medication* and *veterinary visits*, Kendall's tau 0.51, -Log*P* 154.8 0 [*P*<0.001]; *received medication* and *switched to therapeutic diet*, Kendall's tau 0.17, -Log*P* 14.9

**Table 1.** Comparison of summary information from the original dataset of 2,322 owner-dog pairs, and the dataset used for statistical analyses comprising 2,211 owner-dog pairs where the owner was the primary decision-maker.

| Variable [1] | Categories | All owners | | Primary decision-makers | |
|---|---|---|---|---|---|
| | | Number | Percentage | Number | Percentage |
| **Owner variables** | | | | | |
| Location | United Kingdom | 1659 | 71.4% | 1609 | 72.8% |
| | Other European country | 358 | 15.4% | 318 | 14.4% |
| | North America | 128 | 5.5% | 119 | 5.4% |
| | Australia / New Zealand / Oceania | 100 | 3.4% | 97 | 4.4% |
| | Other region | 77 | 3.3% | 68 | 3.1% |
| Setting | Urban | 824 | 35.5% | 780 | 35.3% |
| | Equally urban and rural | 740 | 31.9% | 703 | 31.8% |
| | Rural | 758 | 32.6% | 728 | 32.9% |
| Owner age (years) | <30 | 385 | 16.6% | 340 | 15.4% |
| | 30–39 | 507 | 21.8% | 482 | 21.8% |
| | 40–49 | 474 | 20.4% | 452 | 20.4% |
| | 50–59 | 528 | 22.7% | 515 | 23.3% |
| | ≥60 | 428 | 18.4% | 422 | 19.1% |
| Owner gender | Female | 2146 | 92.4%% | 2055 | 92.9% |
| | Male | 176 | 7.6% | 156 | 7.1% |
| Education | Basic or high school | 439 | 18.9% | 414 | 18.7% |
| | College | 662 | 28.5% | 636 | 28.8% |
| | Graduate | 682 | 29.4% | 645 | 29.2% |
| | Postgraduate | 539 | 23.2% | 516 | 23.3% |
| Income | Low | 364 | 15.7% | 352 | 15.9% |
| | Medium | 1611 | 69.4% | 1530 | 69.2% |
| | High | 347 | 14.9% | 329 | 14.9% |
| Animal-related Career | Yes | 435 | 18.9% | 425 | 19.2% |
| | No | 1887 | 81.3% | 1786 | 80.8% |
| Owner diet [2] | Omnivore | 931 | 40.1% | 901 | 40.7% |
| | Omnivore (restricted) | 500 | 21.5% | 479 | 21.7% |
| | Pescatarian | 124 | 5.3% | 117 | 5.3% |
| | Vegan | 532 | 22.9% | 493 | 22.3% |
| | Vegetarian | 235 | 10.1% | 221 | 10.0% |
| Decision-maker status [3] | Primary | 2211 | 95.2% | 2211 | 100.0% |
| | Some lesser role | 96 | 4.1% | — | — |
| | No role | 15 | 0.06% | — | — |
| **Animal Variables** | | | | | |
| Age (years) | 1 to 5 | 1144 | 49.3% | 1089 | 49.3% |
| | 6 to 20 | 1178 | 50.7% | 1122 | 50.7% |
| Breed size category | | | | | |
| | Medium | 896 | 38.6% | 859 | 38.9% |
| | Toy | 56 | 2.4% | 52 | 2.4% |
| | Small | 467 | 20.1% | 439 | 19.8% |
| | Large | 803 | 34.6% | 764 | 34.5% |
| | Giant | 100 | 4.3% | 97 | 4.4% |
| Sex | Female | 1095 | 47.2% | 1042 | 47.1% |
| | Male | 1227 | 52.8% | 1169 | 52.9% |
| Neuter status | Sexually intact | 512 | 22.0% | 486 | 22.0% |

*(Continued)*

**Table 1.** (Continued)

| Variable [1] | Categories | All owners | | Primary decision-makers | |
|---|---|---|---|---|---|
| | | Number | Percentage | Number | Percentage |
| | Neutered | 1810 | 78.0% | 1725 | 78.0% |
| Dog diet [2] | Conventional | 1230 | 53.0% | 1152 | 52.1% |
| | Raw | 744 | 32.0% | 732 | 33.1% |
| | Vegan | 313 | 13.5% | 295 | 13.3% |
| | Vegetarian | 35 | 1.5% | 32 | 1.5% |
| **Healthcare variables** | | | | | |
| Veterinary visits | None | 408 | 17.6% | 399 | 18.0% |
| | 1 | 907 | 39.1% | 873 | 39.5% |
| | 2 | 506 | 21.8% | 473 | 21.4% |
| | 3 | 200 | 8.6% | 185 | 8.4% |
| | 4 or more | 301 | 13.0% | 281 | 12.7% |
| Received medication | No | 1386 | 59.7% | 1331 | 60.2% |
| | Yes | 936 | 40.3% | 880 | 39.8% |
| Switched to a therapeutic diet | No | 2211 | 95.2% | 2111 | 95.5% |
| | Yes | 111 | 4.8% | 100 | 4.5% |

[1] Definitions of the different categories are given in the original study [15].

[2] Owners were asked to choose a single diet classification based on the main diet ingredients and whether these were conventional, raw, vegetarian or vegan in origin [15]; vegetarian diets could not include meat but could include eggs or milk, whilst vegan diets could not include any [15]

[3] Owners were asked "When choosing your animal's diet, are you the: primary decision-maker, play some lesser role or play no role" [15].

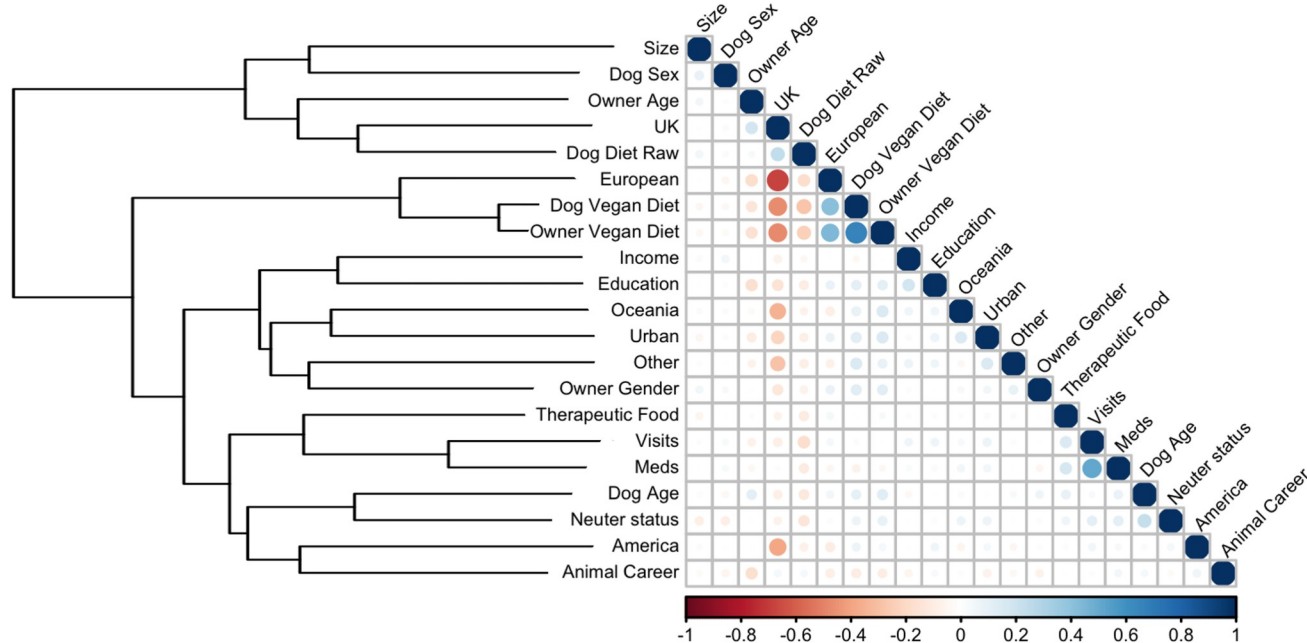

**Fig 2. Naïve correlation amongst owner-animal metadata variables.** Data were first coerced into a numerical ordinal form, except for the Location variable which was not ordinal and, therefore, one-hot encoding was instead used. Correlations between variables were calculated using Kendall's tau, with positive and negative correlations coloured in blue and red, respectively. Both the size and intensity of the colour are proportional to the size of Kendall's tau. Not surprisingly, a large, dark blue circle is evident when variables are correlated with themselves. Hierarchical clustering, using the Ward method [58], is shown on the left side of the figure. All pairs are shown whether statistically significant or not; however, correlation coefficients and associated log *P*-values, corrected for false-discovery rate using the Benjamini-Hochberg procedure, are shown in full in the supplementary tables (S3 and S4 Tables).

0 [*P*<0.001]; *switched to a therapeutic diet* and *veterinary visits*, Kendall's tau 0.16, -Log*P* 15.3 0 [*P*<0.001]). In contrast, weak associations were observed between the *dog vegan diet* variable and the healthcare variables (*dog vegan diet* and *veterinary visits*, Kendall's tau -0.02, -Log*P* 0.5 [*P* = 0.340]; *dog vegan diet* and *received medication*, Kendall's tau -0.11, -Log*P* 2.7 0 [*P* = 0.002]; *dog vegan diet* and *switched to a therapeutic diet*, Kendall's tau 0.00, -Log*P* 0.1 0 [*P* = 0.843]).

## Owner, animal and healthcare variables associated with the *any health problem* binary variable

**Simple regression.** Within the *any health problem* binary variable, 894 (38.5%) and 1429 (61.5%) dogs were classified as 'yes' and 'no', respectively. Simple binary logistic regression was performed first to assess the unadjusted effects of each predictor variable separately using data from primary decision-makers only (Table 2). Broadly similar results were obtained when logistical regression was performed on the dataset from all owners (irrespective of decision-maker status; S1 Table). For example, estimates of the regression coefficients (β) for the *dog diet vegan* variable was -0.43 (SE 0.131) using data from all owners and β -0.30 (SE 0.159) using data from primary decision-makers only.

**Multiple regression.** Next, we created two multiple regression models, again on data from primary decision-makers, with *any health problem* binary as the outcome variable and that either contained all predictor variables or just the owner-animal metadata. For the *all-variable* model (S3A Fig), the variables with the strongest associations with the outcome variable were *dog age* (both the '1-5y' and '6-20y' categories), *received medication*, *veterinary visits* (the '1 visit', '2 visits', '3 visits' and ≥4 visits' categories) and *switched to a therapeutic diet*. In terms of model performance, pseudo-$R^2$ was 0.439, BIC was 1836 and AUC on the test dataset was 0.852 (S5 Fig). For the *owner-animal metadata* model (S3B Fig), the variables with the strongest associations with the outcome variable were *owner age* ('50-59y' and '≥60y' categories), *dog age* ('6-20y' category), *breed size category* ('giant breed' category) and *dog diet* ('vegan diet' category). However, performance of this model was considerably worse in terms of generalisability (BIC 2244), the proportion of variance explained (pseudo-$R^2$ 0.150) and prediction accuracy (AUC on the test dataset: 0.668).

Supervised backwards and forwards stepwise regression was then performed to create a best-fit multiple regression model for the *any health problem* binary outcome variable, starting with the model containing all variables. After refinement by backwards and forwards stepwise elimination, the best-fit model contained 4 variables: *dog age*, *veterinary visits*, *received medication* and *switched to a therapeutic diet* (Fig 3A; model 1, S5 Table). Compared with the all-variable model, generalisability was markedly improved (BIC 1656), with only slight reductions in both the amount of variability explained (pseudo-$R^2$ 0.420) and prediction accuracy (AUC for the test dataset:0.842).

As an additional multiple logistic regression step, the effect of adding the *dog diet vegan* variable to this best-fit model was assessed (Fig 3B; model 2, S4 Table); generalisability was worse (BIC 1660), whilst the amount of variance explained (pseudo-$R^2$ 0.422) and prediction accuracy (AUC for the test dataset 0.845) were similar. Equivalent results were obtained when the *owner diet vegan diet* variable was instead added to the final best-fit model (BIC 1660, pseudo-$R^2$ 0.421, AUC 0.844; Fig 3C; model 2, S4 Table).

## Machine learning predictive modelling with XGBoost

The ability of owner, animal and healthcare variables to predict the *any health problem* variable was further assessed using machine learning predictive modelling using the XGBoost

**Table 2. Results of simple (i.e., univariable) binary logistic regression analyses examining associations between owner, animal and veterinary variables and the *any health problem* binary (the presence of any illness as reported by the owner).**

| Variable [1] | Estimate | Odds ratio | 99%-CI | P-value | Pseudo-R[2] | BIC | AUC |
|---|---|---|---|---|---|---|---|
| **Owner variables** | | | | | | | |
| Location | | | | | 0.0048 | 2231 | 0.491 |
| United Kingdom | Ref | — | — | — | | | |
| Other European country | 0.20 (0.147) | 1.226 | 0.834, 1.787 | 0.165 | | | |
| North America | 0.30 (0.219) | 1.355 | 0.765, 2.373 | 0.165 | | | |
| Australia / New Zealand / Oceania | 0.40 (0.241) | 1.497 | 0.798, 2.783 | 0.094 | | | |
| Other region | 0.23 (0.299) | 1.258 | 0.569, 2.698 | 0.442 | | | |
| Setting [2] | | | | | 0.0034 | 2211 | 0.516 |
| Urban | Ref | — | — | — | | | |
| Rural | 0.22 (0.106) | 1.241 | 0.945, 1.629 | 0.041 | | | |
| Owner age (years) | | | | | 0.0025 | 2234 | 0.556 |
| <30 | Ref | — | — | — | | | |
| 30–39 | -0.01 (0.165) | 0.989 | 0.646, 1.515 | 0.945 | | | |
| 40–49 | -0.04 (1.66) | 0.957 | 0.624, 1.470 | 0.794 | | | |
| 50–59 | -0.24 (0.165) | 0.789 | 0.516, 1.208 | 0.151 | | | |
| ≥60 | -0.12 (0.169) | 0.889 | 0.575, 1.375 | 0.488 | | | |
| Owner gender | | | | | 0.0029 | 2211 | 0.511 |
| Female | Ref | — | — | — | | | |
| Male | -0.39 (0.211) | 0.680 | 0.386, 1.154 | 0.068 | | | |
| Education | | | | | 0.0010 | 2228 | 0.444 |
| Basic or high school | Ref | — | — | — | | | |
| College | 0.00 (0.153) | 1.002 | 0.676, 1.489 | 0.990 | | | |
| Graduate | -0.11 (0.153) | 0.893 | 0.602, 1.326 | 0.458 | | | |
| Postgraduate | 0.02 (0.159) | 1.021 | 0.678, 1.541 | 0.897 | | | |
| Income | | | | | 0.0004 | 2222 | 0.538 |
| Low | Ref | — | — | — | | | |
| Medium | -0.06 (0.140) | 0.941 | 0.657, 1.355 | 0.666 | | | |
| High | -0.13 (0.185) | 0.879 | 0.555, 1.415 | 0.487 | | | |
| Animal-related career | | | | | 0.0005 | 2214 | 0.542 |
| No | Ref | — | — | — | | | |
| Yes | 0.10 (0.125) | 1.108 | 0.800, 1.528 | 0.413 | | | |
| Owner diet [3] | | | | | 0.0089 | 2226 | 0.525 |
| Omnivore | Ref | — | — | — | | | |
| Omnivore (restricted) | 0.21 (0.134) | 1.231 | 0.871, 1.735 | 0.120 | | | |
| Pescatarian | 0.67 (0.223) | 1.947 | 1.094, 3.468 | 0.003 | | | |
| Vegetarian | 0.18 (0.177) | 1.988 | 0.757, 1.883 | 0.305 | | | |
| Vegan | 0.02 (0.137) | 1.013 | 0.710, 1.440 | 0.925 | | | |
| Owner on vegan diet [3] | | | | | 0.0007 | 2214 | 0.503 |
| No | Ref | — | — | — | | | |
| Yes | -0.12 (0.125) | 0.888 | 0.642, 1.221 | 0.341 | | | |
| **Animal Variables** | | | | | | | |
| Age (per year) [4] | | | | | 0.0842 | 2116 | 0.634 |
| 1 to 5 | 0.30 (0.192) | 1.350 | 0.826, 2.229 | 0.119 | | | |
| 6 to 20 | 2.83 (0.285) | 16.943 | 8.199, 35.639 | <0.001 | | | |
| Breed size category | | | | | 0.0041 | 2232 | 0.533 |
| Medium | Ref | — | — | — | | | |

*(Continued)*

**Table 2.** (Continued)

| Variable [1] | Estimate | Odds ratio | 99%-CI | *P*-value | Pseudo-R² | BIC | AUC |
|---|---|---|---|---|---|---|---|
| Toy | 0.04 (0.345) | 1.036 | 0.408, 2.460 | 0.919 | | | |
| Small | 0.30 (0.139) | 1.350 | 0.943, 1.932 | 0.031 | | | |
| Large | 0.14 (0.119) | 1.149 | 0.846, 1.563 | 0.242 | | | |
| Giant | 0.23 (0.251) | 1.261 | 0.652, 2.391 | 0.355 | | | |
| Sex | | | | | 0.0002 | 2215 | 0.471 |
| Female | Ref | — | — | — | | | |
| Male | 0.06 (0.101) | 0.946 | 0.729, 1.228 | 0.585 | | | |
| Neuter status | | | | | 0.0267 | 2182 | 0.522 |
| Sexually intact | Ref | — | — | — | | | |
| Neutered | 0.73 (0.132) | 2.079 | 1.488, 2,943 | <0.001 | | | |
| Dog diet [3] | | | | | 0.0270 | 2196 | 0.536 |
| Conventional | Ref | — | — | — | | | |
| Raw | -0.61 (0.115) | 0.544 | 0.404, 0.729 | <0.001 | | | |
| Vegetarian | 0.08 (0.399) | 1.083 | 0.374, 3.053 | 0.842 | | | |
| Vegan | -0.53 (0.164) | 0.591 | 0.384, 0.897 | 0.001 | | | |
| Dog on vegan diet [2] | | | | | 0.0030 | 2211 | 0.535 |
| No | Ref | — | — | — | | | |
| Yes | -0.30 (0.159) | 0.740 | 0.488, 1.106 | 0.058 | | | |
| **Healthcare variables** | | | | | | | |
| Veterinary visits | | | | | 0.228 | 1933 | 0.770 |
| None | Ref | — | — | — | | | |
| 1 | 0.64 (0.186) | 1.889 | 1.184, 3.053 | <0.001 | | | |
| 2 | 1.94 (0.194) | 6.940 | 4.266, 11.635 | <0.001 | | | |
| 3 | 2.09 (0.234) | 8.064 | 4.413, 15.114 | <0.001 | | | |
| 4 or more | 2.84 (0.232) | 17.098 | 9.555, 31.709 | <0.001 | | | |
| Received medication | | | | | 0.319 | 1772 | 0.768 |
| No | Ref | — | — | — | | | |
| Yes | 2.30 (0.118) | 10.016 | 7.422, 13.622 | <0.001 | | | |
| Switched to therapeutic diet | | | | | 0.0446 | 2160 | 0.542 |
| No | Ref | — | — | — | | | |
| Yes | 1.92 (0.294) | 6,885 | 3.372, 15.657 | <0.001 | | | |

Results presented are from simple (i.e., univariable) binary logistic regression, whereby each independent predictor variable is tested separately in a logistic regression model. These results were then used to determine the variables to include in subsequent multiple regression analysis, as shown in Fig 4 and S4 Table. Results are reported as estimates of regression coefficients (β) with its standard error in brackets, odds ratios and 99% confidence intervals (99%-CI). Model performance assessed using the coefficient of determination (pseudo-R²) based on the method reported by Nagelkerke [63], with pseudo-R², the Bayesian information criterion (BIC [60,61]) and area under the receiver operating characteristic curve (AUC) for the test dataset. For BIC, models having the best fit have lower BIC values; n.b., BIC can only be compared within the same family of models. For AUC, values can range from 0 to 1; a model that performed no better than chance would have an AUC of 0.5, and models predicting better than by chance would have AUC >0.5, with an AUC of 1.0 suggesting perfect prediction

[1] Definitions of the different categories are given in the original study [15]

[2] Please note that the urban category in *setting* variable combines the 'urban' and 'equally urban and rural categories'

[3] Please see the footnote to Table 1 for details of how owner and dog diets were assigned

[4] Dog age analysed as a continuous variable with B-splines, utilising boundary knots and an internal knot at the median value (6 years); therefore, odds ratios represent are the effect per year for each side of that knot.

algorithm. Two separate models were created using data from primary decision-makers; the first contained all variables, whilst the second was a reduced model containing just the owner-animal metadata (i.e., without healthcare variables). Based on ROC analysis, the prediction

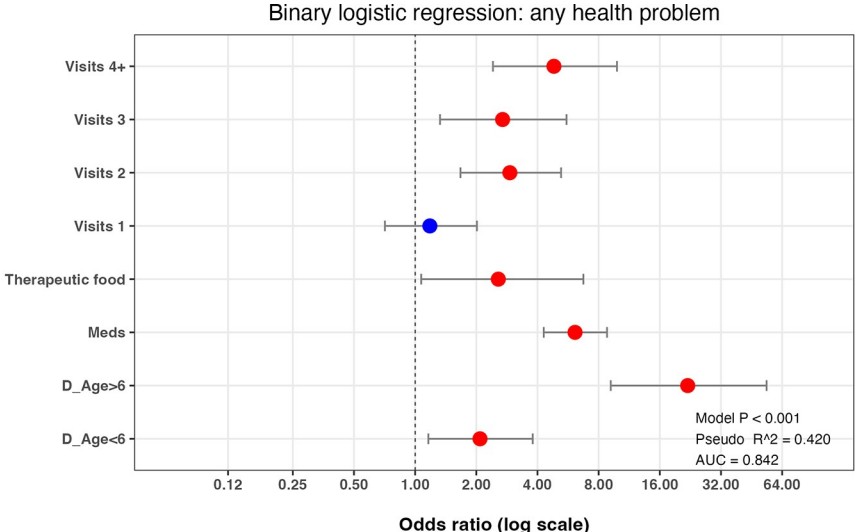

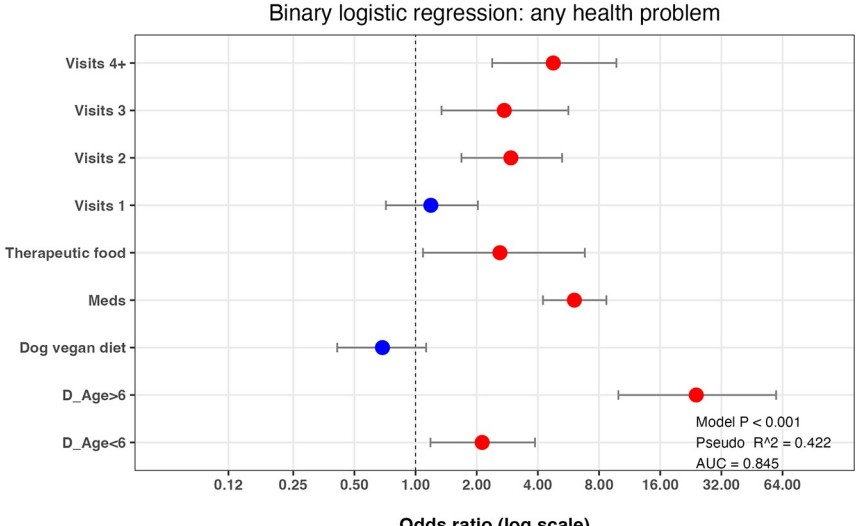

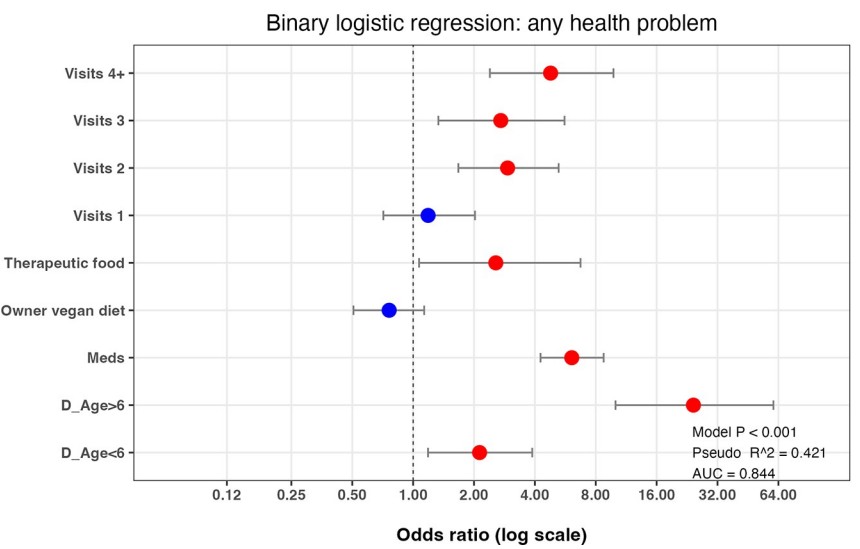

**Fig 3. Final binary logistic regression model of variables associated with the *any health problem* binary in owners who were primary decision-makers.** Results of the best-fit binary logistic regression model of the *any health problem* binary (a) and the effect of adding either the *dog diet vegan* (b) or *owner diet vegan* (c) to this best-fit model. The dots represent the odds ratio for each variable, whilst the bars represent 99% confidence intervals (99%-CI). Variables where the 99%-CI range does not include 1.0 (vertical dotted line) are depicted in red, whilst those that include 1.0 are depicted in blue. Note the logarithmic scale for the X-axis in each graph.

accuracy of the all-variable model was reasonable (AUC 0.836; Fig 4A). The XGBoost procedure also enables the most important variables contributing to the prediction to be identified, for the all-variable model (Fig 4B); the variables that most strongly predicted the *any health problem* outcome variable were (in order, based on the sum of the 3 metrics): *received medication* (importance 0.393, cover 0.129, frequency 0.065), *dog age* (importance 0.175, cover 0.182, frequency 0.188) and *veterinary visits* (importance 0.153, cover 0.181, frequency 0.155), followed by *owner age* (importance 0.048, cover 0.083, frequency 0.106) and then *breed size category* (importance 0.046, cover 0.070, frequency 0.087). Other variables followed these including *education*, *dog sex*, *location*, and *dog diet*; for the latter variable, *dog diet raw* was 10th in order of importance (importance 0.015, cover 0.026, frequency 0.032), whilst *dog diet vegan* was 15th in order of importance (importance 0.008, cover 0.021, frequency 0.013).

The order of importance of predictor variables was broadly similar in the reduced (owner-animal metadata) model although, based on ROC analysis, prediction accuracy was poor (AUC 0.674, Fig 5A). In this reduced model, *dog age* was overwhelmingly most important (importance 0.261, cover 0.196, frequency 0.157), followed by *owner age* (importance 0.115, cover 0.123, frequency 0.141, *education level* (importance 0.095, cover 0.089, frequency 0.124), *breed size category* (importance 0.091, cover 0.099, frequency 0.108) and *neuter status* (importance 0.060, cover 0.062, frequency 0.047). Other variables including *dog diet*, *urban setting*, *location* and *owner income* and followed these (Fig 5B); for the dog diet variable, *dog diet raw*

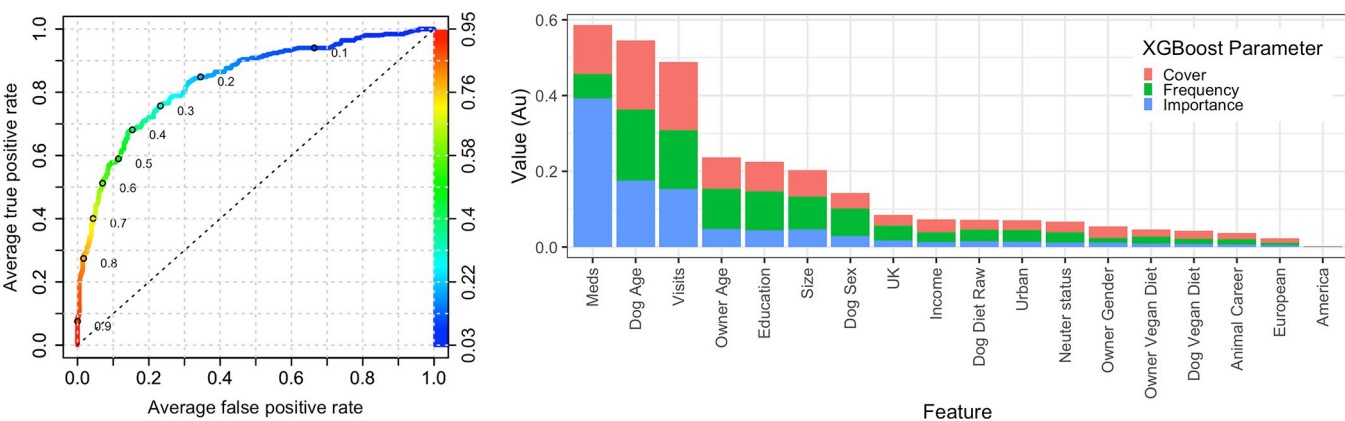

**Fig 4. All-variable XGBoost model on the *any health problem* binary using the primary decision-maker dataset.** (a) Receiver operating characteristic curve of a prediction model containing all variables (owner, animal and healthcare). This shows the increasing true positive and false positive rates, with decrease of the threshold probability for prediction of *any health problem*. Prediction accuracy was good as assessed by ROC analysis (area under curve 0.836, 99%-CI: 0.794–0.878). Acceptance threshold is indicated by the colour bar on the right-hand side and shown at discrete points on the curve. For example, with a threshold probability of 0.2, the model correctly predicts approximately 80% of dogs where the *any health problem* binary was classified as 'yes', but the false-positive rate is approximately 30%. (b) Graph depicting the relative contribution of predictor variables to the all-variable XGBoost model for the *any health problem* binary. Importance' represents fractional contribution of each feature to the model, based on the total gain from including each feature; 'cover' represents the number of observations related to this feature in the model); 'frequency' represents the relative number of times a feature has been used in trees). Variables are organised in order of importance left to right, based on the sum of the 3 metrics, whilst variables that are not shown did not contribute to the final model.

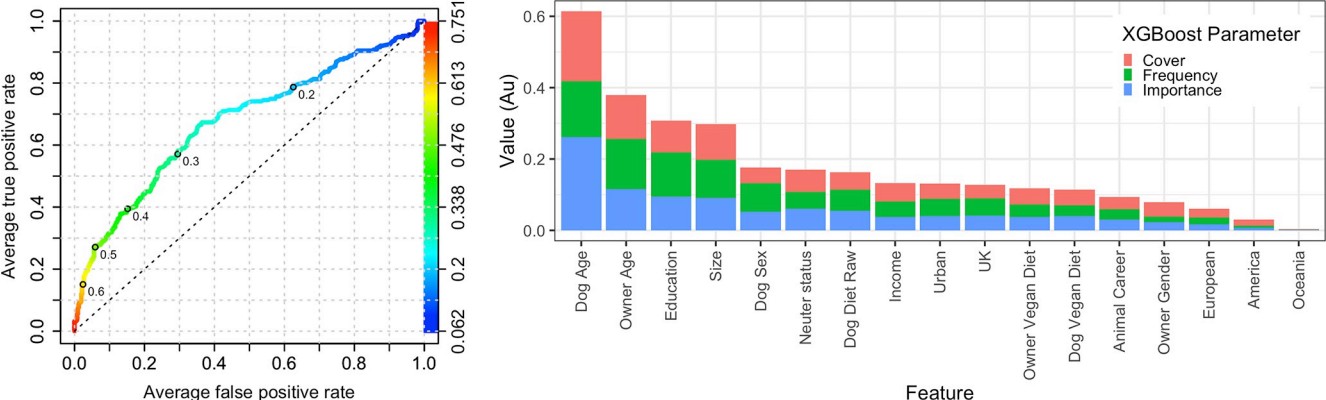

**Fig 5. Reduced (owner-animal metadata) XGBoost model on the *any health problem* binary using the primary decision-maker dataset.** (a) Receiver operating characteristic curve of a reduced prediction model only containing owner and animal variables. This shows the increasing true positive and false positive rates, with decrease of the threshold probability for prediction of *any health problem*. Prediction accuracy was moderate, as assessed by ROC analysis (AUC 0.674, 99%-CI: 0.617–0.731). Acceptance threshold is indicated by the colour bar on the right-hand side and shown at discrete points on the curve. For example, with a threshold probability of 0.2, the model correctly predicts almost 80% of dogs where the *any health problem* binary was classified as 'yes', but the false-positive rate is over 60%. The fact that the prediction thresholds are all low (< 0.5) shows that this model struggles to predict health issues. (b) Graph depicting the relative contribution of predictor variables to the all-variable XGBoost model for the *any health problem* binary. 'Importance' represents fractional contribution of each feature to the model, based on the total gain from including each feature; 'cover' represents the number of observations related to this feature in the model); 'frequency' represents the relative number of times a feature has been used in trees). Variables are organised from left to right, based on the sum of the 3 metrics, whilst variables that are not shown did not contribute to the final model.

was 7th in order of importance (importance 0.054, cover 0.048, frequency 0.060), whilst *dog diet vegan* was 12th in order of importance (importance 0.039, cover 0.045, frequency 0.031).

Results were broadly similar when machine learning predictive modelling with XGBoost was used to create both all-variable and reduced (owner-animal metadata) models for the *any health problem* outcome measure using the all-owner dataset (S5 and S6 Figs).

## Owner, animal and healthcare variables associated with the *significant illness* binary variable

**Simple regression.** Within the *significant illness* binary variable, 122 (5.5%) and 2089 (94.5%) dogs were classified as 'yes' and 'no', respectively. Results of the simple binary logistic regression stage, to assess the unadjusted effects of each predictor variable separately using data from primary decision-makers only, are shown in Table 3, whilst the results of equivalent analyses on the dataset from all owners (irrespective of decision-maker status) are shown in S2 Table. Once again, estimates of the regression coefficients were broadly similar from both data sets. For example, estimates of the regression coefficients (β) for the *dog diet vegan* variable was -0.46 (SE 0.309) using data from all owners and β -0.44 (SE 0.402) using data from primary decision-makers only.

**Multiple regression.** Next, we created both an *all-variable* and an *owner-animal metadata* model, using data from primary decision-makers, with *significant illness* binary as the outcome variable. For the *all-variable* model (S4A Fig), the variables with the strongest associations with the outcome variable were *dog age* ('6-20y' category), *giant breed*, and *received medication*. In terms of model performance, pseudo-$R^2$ was 0.403, BIC was 679 and AUC on the test data-set was 0.890. For the *owner-animal metadata* model (S4B Fig), the variables with the strongest associations with the outcome variable were *dog age* ('6-20y' category) and *giant breed*. Once again, the performance of this model was considerably worse in terms of generalisability (BIC 799), the proportion of variance explained (pseudo-$R^2$ 0.124) and prediction accuracy (AUC on the test dataset: 0.692).

**Table 3. Results of simple (i.e. univariable) binary logistic regression analyses examining associations between owner, animal and healthcare variables and the *significant illness* binary variable ('significant' or 'serious illness' as reported by owners who were primary decision-makers).**

| Variable [1] | Estimate | Odds ratio | 99%-CI | *P*-value | Pseudo-$R^2$ | BIC | AUC |
|---|---|---|---|---|---|---|---|
| **Owner variables** | | | | | | | |
| Location | | | | | 0.0171 | 681 | 0.481 |
| United Kingdom | Ref | — | — | — | | | |
| Other European country | 0.58 (0.297) | 1.787 | 0.788, 3.712 | 0.051 | | | |
| North America | 0.31 (0.483) | 1.368 | 0.310, 4.083 | 0.516 | | | |
| Australia / New Zealand / Oceania | 1.06 (0.402) | 2.884 | 0.901, 7.475 | 0.008 | | | |
| Other region | 0.77 (0.542) | 2.163 | 0.394, 7.311 | 0.155 | | | |
| Setting [2] | | | | | 0.0007 | 667 | 0.516 |
| Rural | Ref | — | — | — | | | |
| Urban | 0.14 (0.234) | 1.153 | 0.621, 2.086 | 0.542 | | | |
| Owner age (years) | | | | | 0.0060 | 687 | 0.488 |
| <30 | Ref | — | — | — | | | |
| 30–39 | 0.43 (0.366) | 1.539 | 0.617, 4.186 | 0.238 | | | |
| 40–49 | 0.07 (0.391) | 1.067 | 0.392, 3.045 | 0.867 | | | |
| 50–59 | 0.16 (0.377) | 1.171 | 0.450, 3.257 | 0.676 | | | |
| ≥60 | -0.18 (0.417) | 0.839 | 0.280, 2.513 | 0.674 | | | |
| Owner gender | | | | | 0.0011 | 667 | 0.501 |
| Female | Ref | — | — | — | | | |
| Male | -0.38 (0.522) | 0.683 | 0.129, 2.145 | 0.466 | | | |
| Education | | | | | 0.0035 | 681 | 0.530 |
| Basic or high school | Ref | — | — | — | | | |
| College | -0.22 (0.360) | 0.803 | 0.319, 2.105 | 0.543 | | | |
| Graduate | 0.79 (0.340) | 1.083 | 0.461, 2.728 | 0.815 | | | |
| Postgraduate | 0.20 (0.348) | 1.220 | 0.506, 3.123 | 0.586 | | | |
| Income | | | | | 0.0001 | 675 | 0.514 |
| Low | Ref | — | — | — | | | |
| Medium | 0.01 (0.314) | 1.009 | 0.473, 2.447 | 0.977 | | | |
| High | -0.07 (0.420) | 0.932 | 0.304, 2.771 | 0.867 | | | |
| Animal-related career | | | | | 0.0041 | 666 | 0.533 |
| No | Ref | — | — | — | | | |
| Yes | 0.40 (0.257) | 1.485 | 0.739, 2.811 | 0.124 | | | |
| Owner diet [3] | | | | | 0.0068 | 686 | 0.578 |
| Omnivore | Ref | — | — | — | | | |
| Omnivore (restricted) | 0.35 (0.296) | 1.417 | 0.647, 3.026 | 0.239 | | | |
| Pescatarian | -0.26 (0.618) | 0.774 | 0.102, 2.994 | 0.678 | | | |
| Vegetarian | 0.57 (0.356) | 1.774 | 0.664, 4.287 | 0.108 | | | |
| Vegan | 0.22 (0.310) | 1.247 | 0.545, 2.736 | 0.475 | | | |
| Owner on vegan diet [3] | | | | | 0.000 | 668 | 0.485 |
| No | Ref | — | — | — | | | |
| Yes | 0.04 (0.274) | 1.042 | 0.490, 2.036 | 0.881 | | | |
| **Animal Variables** | | | | | | | |
| Age (per year) [4] | | | | | 0.0592 | 643 | 0.704 |
| 1 to 5 | 0.10 (0.382) | 1.104 | 0.333, 4.239 | 0.840 | | | |
| 6 to 20 | 2.98 (0.527) | 19.652 | 5.124, 79.645 | <0.001 | | | |
| Giant breed [5] | | | | | 0.0081 | 663 | 0.505 |
| No | Ref | — | — | — | | | |

*(Continued)*

**Table 3.** (Continued)

| Variable [1] | Estimate | Odds ratio | 99%-CI | *P*-value | Pseudo-R$^2$ | BIC | AUC |
|---|---|---|---|---|---|---|---|
| Yes | 0.91 (0.393) | 2.473 | 0.786, 6.223 | 0.021 | | | |
| Sex | | | | | 0.0023 | 667 | 0.498 |
| Female | Ref | — | — | — | | | |
| Male | -0.25 (0.227) | 0.778 | 0.430, 1.340 | 0.270 | | | |
| Neuter status | | | | | 0.0081 | 663 | 0.546 |
| Sexually intact | Ref | — | — | — | | | |
| Neutered | 0.64 (0.330) | 1.904 | 0.873, 4.906 | 0.051 | | | |
| Dog diet [3] | | | | | 0.0124 | 676 | 0.596 |
| Conventional | Ref | — | — | — | | | |
| Raw | -0.58 (0.268) | 0.559 | 0.269, 1.086 | 0.030 | | | |
| Vegetarian | 0.24 (0.750) | 1.269 | 0.091, 6.167 | 0.751 | | | |
| Vegan | -0.63 (0.410) | 0.533 | 0.157, 1.375 | 0.125 | | | |
| Dog on vegan diet [2] | | | | | 0.0025 | 666 | 0.518 |
| No | Ref | — | — | — | | | |
| Yes | -0.44 (0.402) | 0.642 | 0.192, 1.615 | 0.271 | | | |
| **Healthcare variables** | | | | | | | |
| Veterinary visits | | | | | 0.2417 | 554 | 0.841 |
| None | Ref | — | — | — | | | |
| 1 | 0.31 (0.670) | 1.358 | 0.279, 11.118 | 0.648 | | | |
| 2 | 1.30 (0.645) | 3.667 | 0.842, 28.960 | 0.044 | | | |
| 3 | 1.68 (0.698) | 5.372 | 0.983, 45.803 | 0.016 | | | |
| 4 or more | 3.54 (0.603) | 34.615 | 9.942, 256.974 | <0.001 | | | |
| Received medication | | | | | 0.197 | 558 | 0.777 |
| No | Ref | — | — | — | | | |
| Yes | 3.09 (0.427) | 21.931 | 8.374, 81.002 | <0.001 | | | |
| Switched to therapeutic diet | | | | | 0.050 | 640 | 0.555 |
| No | Ref | — | — | — | | | |
| Yes | 1.87 (0.310) | 6.460 | 2.773, 13.900 | <0.001 | | | |

Results presented are from simple (i.e., univariable) binary logistic regression, whereby each independent predictor variable is tested separately in a logistic regression model. These results were then used to determine the variables to include in subsequent multiple regression analysis, as shown in Fig 4 and S4 Table. Results are reported as estimates of regression coefficients (β) with its standard error in brackets, odds ratios and 99% confidence intervals (99%-CI). Model performance assessed using the coefficient of determination (pseudo-R$^2$) based on the method reported by Nagelkerke [63], with pseudo-R$^2$, the Bayesian information criterion (BIC [60,61]) and area under the receiver operating characteristic curve (AUC) for the test dataset. For BIC, models having the best fit have lower BIC values; n.b., BIC can only be compared within the same family of models. For AUC, values can range from 0 to 1; a model that performed no better than chance would have an AUC of 0.5, and models predicting better than by chance would have AUC >0.5, with an AUC of 1.0 suggesting perfect prediction

[1] Definitions of the different categories are given in the original study [15]

[2] Please note that the urban category in *setting* variable combines the 'urban' and 'equally urban and rural categories'

[3] Please see the footnote to Table 1 for details of how owner and dog diets were assigned

[4] Dog age analysed as a continuous variable with B-splines, utilising boundary knots and an internal knot at the median value (6 years); therefore, odds ratios represent are the effect per year for each side of that knot

[5] Please note that breed was better categorised using the *breed size category* variable for regression models using the *significant illness* outcome.

Supervised backwards and forwards stepwise logistic regression was again used to create a best-fit multiple regression model for the *significant illness* outcome variable, starting with all variables. After refinement by backwards and forwards stepwise elimination, the best-fit model contained 3 variables: *dog age*, *veterinary visits* and *received medication* (Fig 6A; model 1, S5 Table). Compared with the *all-variable* model, generalisability was markedly improved

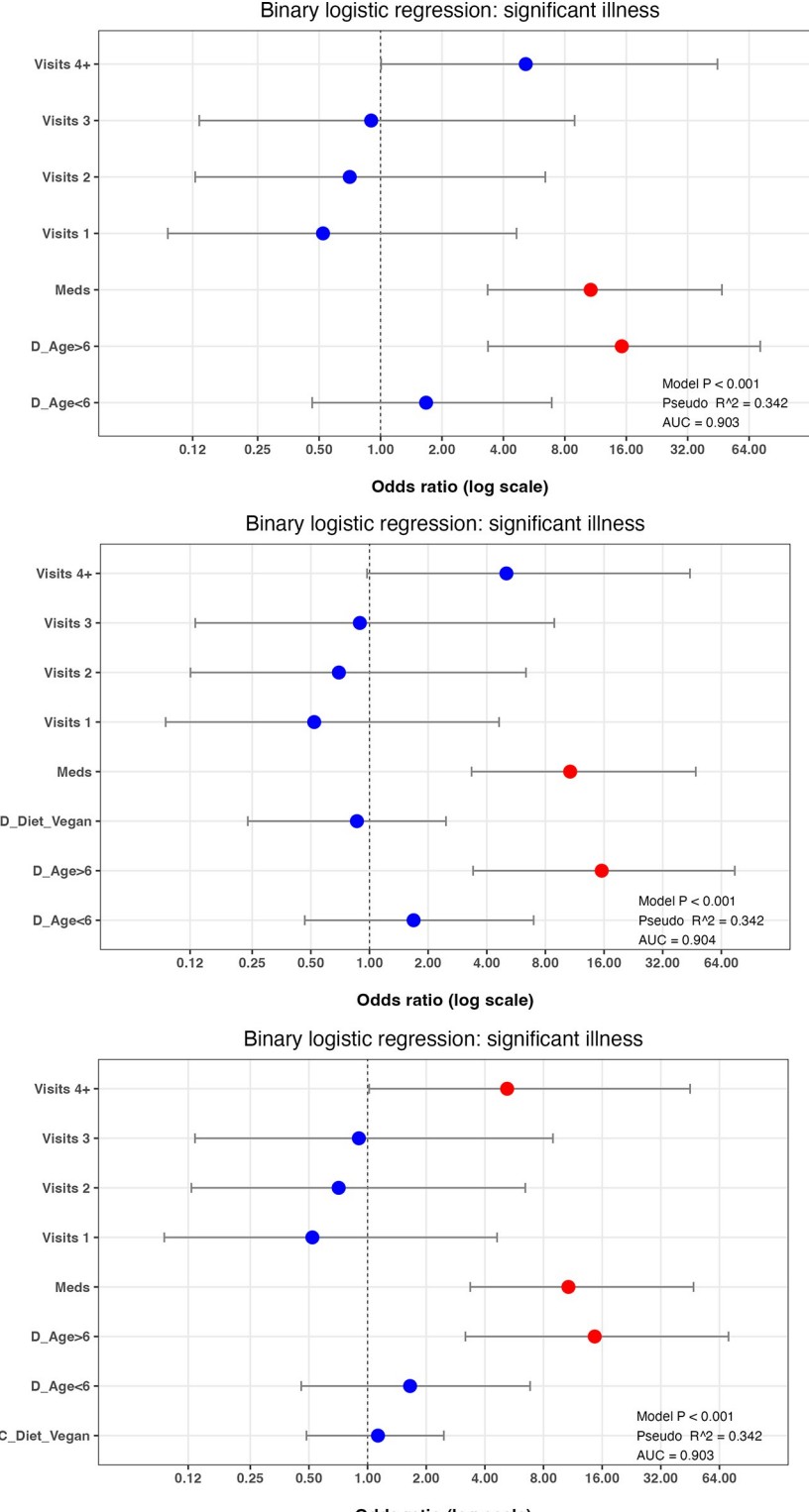

**Fig 6. Final best-fit binary logistic regression model of variables associated with the *significant illness* binary in owners who were primary decision-makers.** Results of the best-fit binary logistic regression model of the *significant illness* binary (a) and the effect of adding either the *dog diet vegan* (b) or *owner diet vegan* (c) to this best-fit model. The dots represent the odds ratio for each variable, whilst the bars represent 99% confidence intervals (99%-CI). Variables where the 99%-CI range does not include 1.0 (vertical dotted line) are depicted in red, whilst those that include 1.0 are depicted in blue. Note the logarithmic scale for the X-axis in each graph.

BIC 517), with better prediction accuracy (AUC for the test dataset: 0.903), and only a small decreased in the amount of variability explained (pseudo-$R^2$ 0.342).

As a further multiple logistic regression step, the effect of adding the *dog diet vegan diet* variable to this best-fit model was assessed (Fig 6B; model 2; S6 Table); generalisability was worse (BIC 524), whilst the amount of variance explained (pseudo-$R^2$ 0.342) and prediction accuracy (AUC for the test dataset 0.904) were similar. Equivalent results were obtained when the *owner diet vegan diet* variable was instead added to the final best-fit model (BIC 524, pseudo-$R^2$ 0.342, AUC 0.903; Fig 6C; model 2; S6 Table).

**Machine learning predictive modelling with XGBoost.** Using machine learning predictive modelling with XGBoost, and data from primary decision-makers, all-variable and reduced (owner-animal metadata) models were created (S2 File) for the *serious illness* outcome variable. Based on ROC analysis, the prediction accuracy of the all-variable model was reasonable (AUC 0.887; Fig 7A). In order (Fig 7B), the variables that most strongly predicted the *significant illness* outcome variable were: *veterinary visits* (importance 0.530, cover 0.473, frequency 0.285), *dog age* (importance 0.182, cover 0.219, frequency 0.302) and *received medication* (importance 0.175, cover 0.190, frequency 0.143), followed by *breed size category* (importance 0.050, cover 0.034, frequency 0.079), *education* (importance 0.016, cover 0.020, frequency 0.050), *owner age* (importance 0.012, cover 0.014, frequency 0.036) and *dog sex* (importance 0.009, cover 0.010, frequency 0.034). Other variables followed these including *location*, *urban setting*, *owner sex*, *animal career* and *therapeutic food*. For the *dog diet* variable, *dog diet raw* was 17th in order of importance (importance 6.14 x10$^{-5}$, cover 4.09 x10$^{-5}$, frequency 1.55 x10$^{-4}$) relative importance), whilst the *dog diet vegan* variable was of insufficient importance to be included in the model (i.e., relative importance less than that of the *dog diet raw*).

As with the *any health problem* binary, ROC analysis suggested that prediction accuracy of the reduced (owner-animal metadata) model was poor (AUC 0.689, Fig 8A). Once again, *dog age* (importance 0.304, cover 0.362, frequency 0.224) was the most important predictor,

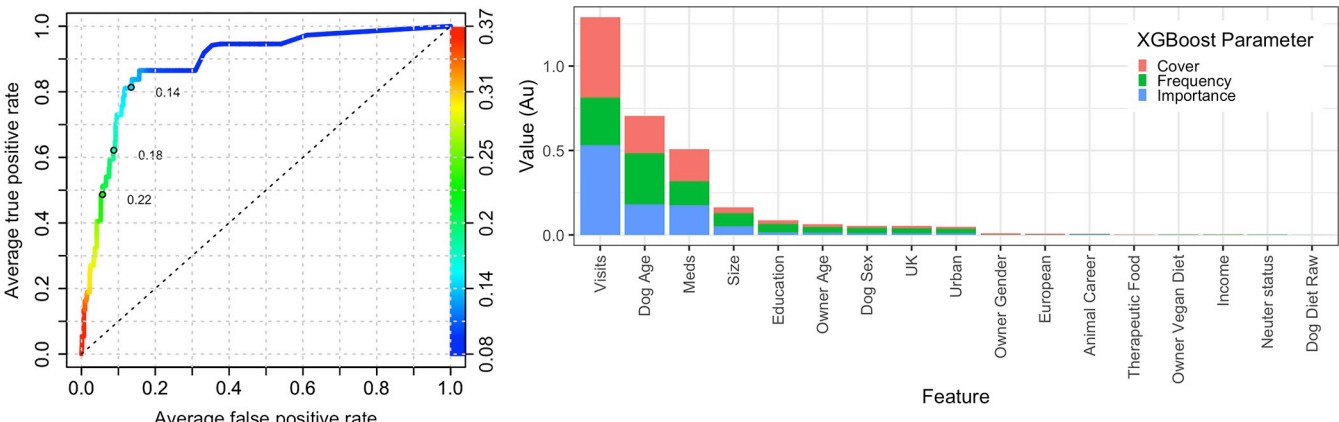

**Fig 7. All-variable XGBoost model on the 'significant illness' binary using the primary decision-maker dataset.** (a) Receiver operating characteristic curve of a prediction model containing all variables (owner, animal and healthcare). This shows the increasing true positive and false positive rates, with decrease of the threshold probability for prediction of *any health problem*. Prediction accuracy was good, as assessed by ROC analysis (area under curve 0.887, 99%-CI: 0.816–0.959). Acceptance threshold is indicated by the colour bar on the right-hand side and shown at discrete points on the curve. For example, with a threshold probability of 0.14, the model correctly predicts over 80% of dogs where the *any health problem* outcome was classified as 'yes', and the false-positive rate is less than 15%. (b) Graph depicting the relative contribution of predictor variables to the all-variable XGBoost model for the *any health problem* binary. 'Importance' represents fractional contribution of each feature to the model, based on the total gain from including each feature; 'cover' represents the number of observations related to this feature in the model); 'frequency' represents the relative number of times a feature has been used in trees). Variables are organised from left to right, based on the sum of the 3 metrics, whilst variables that are not shown did not contribute to the final model.

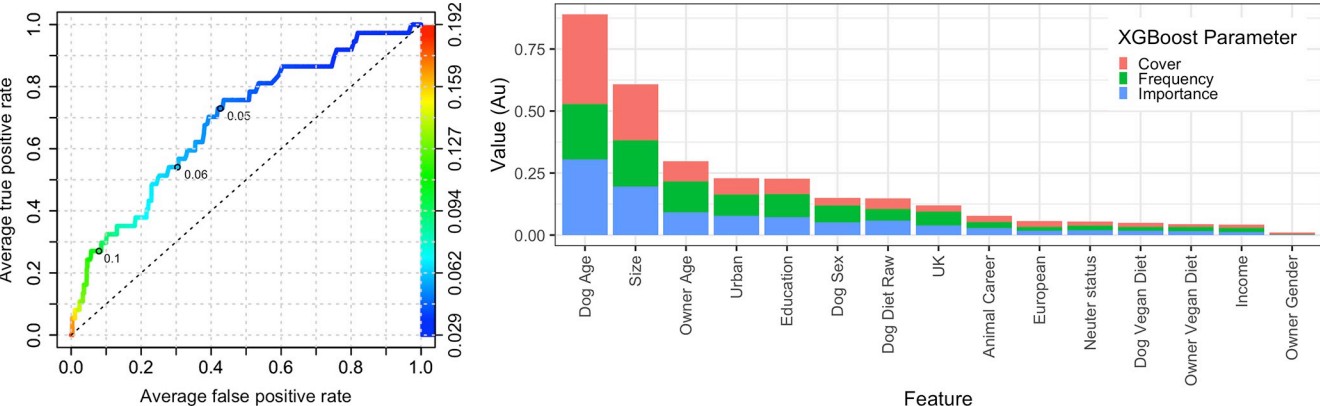

**Fig 8. Reduced (owner-animal metadata) XGBoost model on the *significant illness* binary using the primary decision-maker dataset.** (a) Receiver operating characteristic curve of a reduced prediction model only containing owner and animal variables. This shows the increasing true positive and false positive rates, with decrease of the threshold probability for prediction of *significant illness*. Prediction accuracy was moderate, as assessed by ROC analysis (area under curve 0.689, 99%-CI: 0.572–0.805). Acceptance threshold is indicated by the colour bar on the right-hand side, and also shown at discrete points on the curve. For example, with a threshold probability of 0.05, the model correctly predicts approximately 73% of dogs where the *significant illness* outcome was classified as 'yes', but the false-positive rate is approximately 45%. (b) Graph depicting the relative contribution of predictor variables to the all-variable XGBoost model for the *any health problem* binary. Importance' represents fractional contribution of each feature to the model, based on the total gain from including each feature; 'cover' represents the number of observations related to this feature in the model); 'frequency' represents the relative number of times a feature has been used in trees). Variables are organised from left to right, based on the sum of the 3 metrics, whilst variables that are not shown did not contribute to the final model.

followed by *breed size category* (importance 0.195, cover 0.226, frequency 0.186), *owner age* (importance 0.091, cover 0.082, frequency 0.125), *urban setting* (importance 0.077, cover 0.066, frequency 0.086) and *education* (importance 0.072, cover 0.060, frequency 0.094); other variables followed these including *dog sex*, *dog diet*, *location*, *animal career* and *neuter status* (Fig 8B); for the dog diet variable, *dog diet raw* was 7th in order of importance (importance 0.059, cover 0.042, frequency 0.047), whilst *dog diet vegan* was 12th in order of importance (importance 0.018, cover 0.014, frequency 0.016).

Results were broadly similar when machine learning predictive modelling with XGBoost was used to create both all-variable and reduced (owner-animal metadata) models for the *significant illness* outcome variable using the all-owner dataset (S7 and S8 Figs).

## Discussion

In the current study, we examined associations between owner perceptions of the health of their dog and a range of owner-related, animal-related and healthcare variables. Predictor variables that were most strongly associated with our outcome variables were dog age and healthcare variables (including number of veterinary visits and receiving medication), whilst the contribution from other variables was more limited, and the effect of dog diet was negligible. We utilised data from a previous study, where the opinions of owners about the health of their dog was gathered by questionnaire [15]. One of the aims of that study was to examine associations between the type of diet that owners predominantly fed, and both subjective owner opinions about their dog's health as well as their recollection of other health information (e.g., medication usage and the presence of specific health disorders). A key conclusion of that study was that health status was positively associated with feeding either a vegan or raw diet, compared with other diet types (including conventional and vegetarian). Given the focus of that study, other information gathered in the questionnaire was not assessed and no account was taken of possible confounding amongst variables. Therefore, to extend the findings of the previous work, additional owner, animal and healthcare variables were studied. Previous work

has reported associations between such variables and owner decisions about feeding. For example, animal variables, such as age and neuter status, were associated with the owner feeding choice in the previous study of Knight et al. [15]. In the same study, maintenance of pet health was the most-common reason cited by owners for choosing a particular food [15] whilst, in other research, owner characteristics (such as geographic location) are also associated with food choice [65]. Finally, owner characteristics can also be associated with aspects of dog health; for example, both owner age and income are associated with the prevalence of obesity in dogs [66]. Considering these previous findings, our decision to examine variables beyond those of diet type was justified.

To ensure that owner perceptions of dog health were examined, we created two outcome variables from a question in the original survey question where owners were asked to rate their dog's health over the previous 12 months [15]. Such responses are not objective measures of health; rather, they require owners to formulate a subjective holistic opinion, which will necessarily require them to take many factors into account (Fig 9A). There will be a contribution from the actual health of their dog (which in turn is associated with other factors), but this will be influenced by their knowledge and recollection, whereas general opinions on what constitutes 'health' will be affected by owner attitudes, beliefs and behaviours.

This study also differed from the original study because we excluded dogs <1y because numbers were small (n = 26) and, arguably, insufficient to reveal insights into how the perceptions of owners with dogs in their growth phase might vary. We were also concerned about the small numbers (n = 111) of respondents who were not make the primary decision about their dog's diet. Some of these owners reported playing "some lesser role", whilst others reported playing "no role" in diet decision-making [15]. Not only would this raise concerns about the accuracy of the diet information, but these owners might also be less involved in other aspects of the dog's life, including health and veterinary matters, which might adversely affect the accuracy of other information provided (Fig 9A). There might also be differences in their opinions and attitudes towards health matters in general. Analyses performed using *decision-maker status* as an outcome variable supported these concerns, not least since the odds of a dog being recorded as having a health problem were less for owners who were primary decision-makers than for those who were not. One way to account for this would be to include *decision-maker status* as a covariate in all analyses but the marked imbalance in group size (2,211 vs. 111) could adversely affect the performance of procedures such as logistic regression. Therefore, we instead decided to limit further analyses to owners who were primary decision-makers only. Decisions about excluding observations from a dataset should be made with caution, properly justified and consider the possible impact on results. Based upon our flow diagram of associations within in the dataset (Fig 9B), we determined that it would be possible to eliminate all pathways associated with the *decision-maker status* variable, thereby simplifying the overall structure without affecting any other associations amongst variables. This was because removing all data from owners who are not primary decision-makers effectively 'collapses' the *decision-maker status* variable into a single category such that variance is no longer associated with it. Estimates of regression coefficients and odds ratios were broadly similar in logistic regression models using the all-owner dataset (S1 and S2 Tables), compared with models using the primary decision-maker dataset (Tables 2 and 3). Any differences are likely to be the result of differences in the characteristics, attitudes and beliefs of the owners that were not primary decision-makers. Of course, one main disadvantage of focusing on data from owners who are primary decision-makers is that it limits the generalisability of findings to the wider pet-owning public, not least to other members of a dog-owning family. However, given the original concern about data reliability, further studies involving all family members would be required to explore such opinions properly.

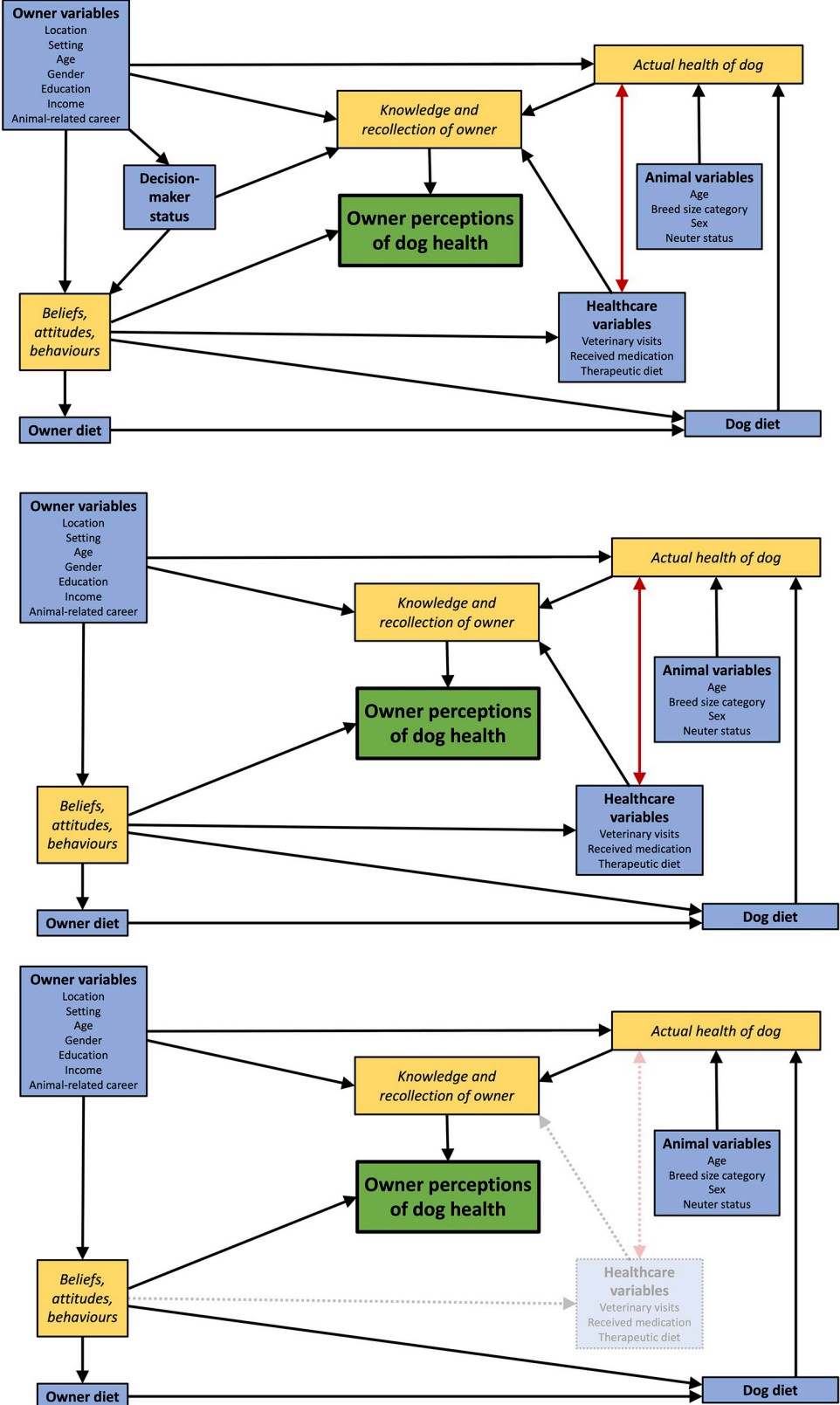

**Fig 9. Flow diagrams illustrating possible pathways for associations between owner perceptions of dog health and the owner, animal and healthcare variables.** The green box depicts the outcome variable (owner perception of

health), blue boxes depict the predictor variables and yellow boxes depict possible mechanisms through which predictor variables may act. Arrows represent the most likely direction of associations, which are unidirectional except for associations between actual dog health and healthcare variables (red double-headed arrow), because these might plausibly act in both directions. (a) Possible associations amongst all available variables within the all-owner dataset; as illustrated, owner perceptions of health will be associated with the actual health of the dog, but the effect could be modified by owner knowledge and recollection, as well as being influenced by attitudes, beliefs and behaviours; decision-maker status has the potential to influence both of these. (b) Possible associations amongst all available variables for the primary decision-maker dataset, where any variance from the decision-maker variable is eliminated by removal of owners who are not primary decision-makers; this has the effect of simplifying possible pathways through which the variables act. (c) Possible associations amongst variables in models on the primary decision-maker dataset where the healthcare variables are not included; as illustrated, the impact of this is different from (b) because no data or variance are removed; therefore, although the healthcare variables are ignored in the modelling, the variance associated within them remains within the dataset, with the potential to affect performance.

The approach of the current study also differed by our decision to include healthcare variables (*received medication*, *veterinary visits*, *switched to a therapeutic diet*) as predictors. In the previous study [15], these variables were instead used as proxy measures for dog health and compared amongst dogs fed different diet types. Whilst both owner perceptions of health and healthcare variables will be associated with actual dog health, they are materially different. As already discussed, owner perceptions of health were derived from a question where owners subjectively rated their dog's health [15], and these perceptions will be influenced by other factors including owner knowledge and recollections, as well as their beliefs and attitudes (Fig 9). Conversely, the healthcare variables were derived from survey questions that required factual recall rather than the formulation of a subjective, holistic opinion about health. For example, owners were asked whether their dog "had received medications in the last year" and how many times the dog had "visited a veterinarian or received a home visit" [15]. Given that such healthcare information will contribute to the knowledge an owner utilises when constructing opinions about the health of their dog, we reasoned that they were valid predictor variables in our statistical modelling. Further, excluding them from statistical models could be problematic as illustrated in Fig 9C; unlike removing observations from owners who were not primary decision-makers (as discussed above), where any variance associated with the variable is removed, simply excluding a variable from a statistical model will not eliminate any associated variance. The 'hidden' variance associated with the healthcare variables might then be incorrectly attributed to other variables leading to erroneous estimates of their regression coefficients. Any unassigned variance also adversely affects model performance, which likely explains why the models that only included owner-animal metadata performed poorly as discussed below.

Using simple logistic regression, we identified associations between owner perceptions of dog health and several owner (e.g., *location*, *diet*), animal (e.g., *age*, *breed*, *neuter status*) and healthcare (e.g., *veterinary visits*, *received medication*, *switched to a therapeutic food*) variables. Further, owner perceptions of dog health were associated with the *dog diet* variable in simple logistic regression (both *raw diet* and *vegan diet* associated with the *any health problem* variable; *raw diet* associated with the *serious illness* variable). These findings are not surprising, and would be expected because they were the result of univariable analyses (simple logistic regression) and, therefore, were broadly similar to the univariable analyses (e.g., odds ratios, one-way ANOVA) conducted in the previous study [15]. In the current study, and as described in the methods section, data pre-processing was necessary to ensure that our statistical analyses were valid. The approach taken was to remove or combine groups where numbers of dogs was small, to ensure better balance amongst categories within a predictor variable. The fact that the results obtained from our univariable analyses (simple logistic regression) were similar to those of Knight et al. [15] suggests that, despite this pre-processing, the final dataset remained representative of the dataset from which it was drawn.

We next used a combination of multiple logistic regression and machine learning predictive modelling so that multiple variables could be analysed concurrently. This enabled the creation of models that best predicted owner perceptions of dog health, and to determine the relative importance of variables contributing to the final models. We chose this combined-methods approach to maximise the benefits of each whilst minimising their disadvantages. Multiple logistic regression is a well-known, widely available technique that is familiar to most people and has been extensively used in veterinary studies including those in dogs [67–70]. Procedures for supervised model selection are well established, such as those based on BIC [60,61], which can reduce the risk of model overfitting and can take relevant prior knowledge into account. For example, in our exploratory analyses, we tested interactions with possible clinical (e.g., between the *veterinary visit* and *received medication* variables), biological (between the *dog sex* and *neuter status* variables), epidemiological (between the *location* and *setting* variables) and psychological (between *owner diet* and *dog diet* variables) explanations. One disadvantage of our manual selection method is that it could introduce bias into the selection process. Other disadvantages include the fact that logistic regression makes certain statistical assumptions (e.g., independence of errors, the requirement for a linear relationship between the logit of the outcome and predictors, absence of multicollinearity and lack of strong influential outliers; [71]) which, if not met, could lead to poor prediction accuracy. Key advantages of machine learning methods are the fact that they make fewer prior assumptions about data distribution and, therefore, can better model complex, nonlinear relationships between the outcome and predictors even when there is significant noise [72]. However, there are disadvantages including requiring a large dataset, the time and resources needed for computation, challenges with interpretation of the models produced and susceptibility to errors [73]. In this respect, any machine learning procedure is susceptible to overfitting, although this depends upon context; models are often intended to be generalisable to a wide range of datasets beyond the dataset that they were trained on. Arguably, this would be less of an issue for the current study because many of the variables of significance were identified using both multiple logistic regression and machine learning predictive modelling, whilst their relative contribution to model prediction were broadly similar (see below). Therefore, despite the limitations of the different modelling techniques, our findings should be valid.

Using multiple logistic regression, the main variables associated with dog health in the best-fit models were *dog age*, *veterinary visits*, and whether the dog had been prescribed medication; in some models, whether the dog had been switched to a therapeutic food was also of significance, especially when owners described their dogs as "generally healthy, but with minor or infrequent problems" (e.g., binary logistic regression on the *any health problem* variable). Depending on the model used, these variables explained between 34% and 52% of variance within the dataset, with very little additional variance (2% to 6%, depending on the model) gained by the addition of any or all other predictor variables. Many of the same variables were also identified using XGBoost, and prediction accuracy (as measured by AUC) was similar, but other predictor variables were also identified including *owner age* and *education*. However, the diet of the dog had little to no effect in all models, except for those that only included owner-animal metadata, albeit that these models performed poorly. Therefore, we could not find convincing evidence that particular food types (conventional, vegan, vegetarian or raw) have any clinically-meaningful positive or negative association with owner opinions of dog health status. As with the original study [15], these analyses were limited by the dataset used, including its cross-sectional design, meaning that causality of associations cannot be assumed, and its reliance on subjective owner perceptions of health status, which might not reflect the actual health of the dogs. Future studies should be considered, for example, cohort studies or

randomised trials utilising objective measures of health such as information from electronic patient record as or formal diagnoses of disease by a veterinarian.

A key limitation of using questionnaires to ascertain the perceptions of owners about dog health is that such information might be affected by owner recollection leading to errors in classification of disease status, so-called response bias [74]. Responses might be biased for various reasons including acquiescence, courtesy, the order of questions or even the perceived social desirability of the responses [74]. Recall bias is a form of response bias whereby the recollections of the owner are affected by their past experiences [75]. Owners of dogs with health problems might search their memories more thoroughly for possible risk factors (such as diet choice, number of visits to the vet, use of medication etc, diet changes) than owners of dogs that do not have health problems, with the potential to exaggerate any associations identified.

As discussed above, it is also feasible that the attitudes and beliefs of owners might either consciously or unconsciously have influenced responses about the health of their dog, a point that the authors of the previous study emphasised [15]. For example, owners who believe a particular type of diet to be optimal for dog health, might be more likely to perceive their dog to be healthy, whether or not this was objectively true. Whilst such a bias would feasibly affect any diet type, a greater bias might be expected with diets perceived to be 'unconventional', again, as acknowledged in the previous study [15]. Although increasing in popularity [1], both vegan diets and raw meat diets are still uncommon choices for owners, and many veterinary professionals either do not recommend them or might even advise against their use. Given that owners who make an active choice to feed either a raw or vegan diet might be acting contrary to veterinary advice, they would arguably be more defensive about this decision than would an owner who feeds a conventional diet. This in turn might unconsciously have biased responses about the health of their dog, for example, minimising the significance of any illnesses. In the original study questionnaire, the authors did attempt to minimise the influence of canine diet variables on reported health status, by placing the health-related questions before most other variables [15]. However, it is unclear as to whether this would adequately eliminate pre-existing unconscious biases resulting from diet choice.

One variable that featured consistently in all models was dog age, whereby illness severity worsened as age increased, especially when in dogs that were ≥6 years. Perceptions of health in senior dogs are likely to be associated with quality of life, which is known to be negatively associated with age [76]. However, it might also be because many chronic diseases have associations with increasing age, including osteoarthritis [68], obesity [77], diabetes mellitus [70] and many forms of cancer [78]. It is noteworthy that such chronic diseases are also negatively associated with quality of life [79–82]. An alternative explanation for the negative association between age and health in the current study is the fact that owners of older dogs perceive their health to be poorer simply because they expect older dogs to be less healthy [83].

Owner perceptions about dog health were also positively associated with them reporting that their dog had received medication in the last 12 months. Once again, the observational study design means that causality cannot be assumed. It is unclear whether such dogs are receiving these medications because they are genuinely less healthy (which is why they were prescribed), whether the knowledge that their dog is receiving medication means the owner perceives their dog to be less healthy or both (Fig 9). Finally, the association might be indirect, if both the outcome and predictor variable were associated with a third variable. In this regard, there was also a strong positive association between owner perception of health and the frequency of veterinary visits, whilst a positive correlation between frequency of veterinary visits and receiving medication was also evident, meaning that all three variables are likely to be inter-related. Further, in some models, another healthcare variable (switching to a therapeutic diet) potentially associated with frequency of veterinary visits was also included. Besides these,

several additional associations were identified with XGBoost, but not with logistic regression, the most important of which were *owner age* and *education*. Owners that were older and those with a higher level of education perceived their dogs to be healthier than did other owners. Further studies would be required to explore all such associations further and determine their true significance to canine health.

Although the study was large, the population studied was not representative of the typical dog-owning public. In this respect, 33% and 13% of owners reportedly fed their dog raw and vegan diets, respectively, which is greater than expected; for example, in a recent UK survey, the proportion of owners feeding raw and vegan diets were 7% and <1%, respectively [1]. Further, the proportion of the UK human population reportedly consuming a vegan diet is estimated to be between 2 and 3% [2] whilst, in the current study, 23% of owners reported consuming a vegan diet. This raises a concern about how representative the study population was, which is of particular significance given the strong clustering based on owner diet. Although the reasons for such an imbalance is not clear, it might well have resulted from the method of owner recruitment. In this respect, the study was widely advertised via social media but, given concerns that owners feeding dogs unconventional diets might be under-represented, relevant interest groups were actively targeted and invited to participate. This strategy was successful in ensuring that adequate numbers of dog owners feeding unconventional diets were included, ensuring that statistical comparisons were meaningful in both the previous [15] and current studies. However, the recruitment strategy might have inadvertently generated an unrepresentative study population which, in turn, may have influenced the results obtained, as discussed above.

In addition to the unrepresentative study population, multiple associations amongst groups were identified with the potential for confounding amongst variables. For example, feeding a raw diet was more commonly reported by owners from the UK, whilst feeding a vegan diet was more common in owners from Europe. Further, there were positive associations between owner income and education, and between dog age and neuter status. Including healthcare variables in multiple logistic regression analyses might also have hidden the associations seen with the *owner-animal metadata*, including diet type. In logistic regression analyses, we accounted for these associations by determining multicollinearity using VIF, refining models where this might be a problem. No significant issues of multi-collinearity amongst predictor variables were identified in any of the final models. Whilst this does not necessarily guarantee that variance from each factor in the models was correctly assigned, any errors are likely to be small. The use of XGBoost would further address these concerns given that these techniques do not make prior assumptions about data distribution and can better take account of complexity in interactions amongst variables [69]. Nonetheless, to ensure any *owner-animal metadata* effects were not unfairly penalised by the inclusion of the healthcare variables, we also examined models containing only the *owner-animal metadata*. In these models, *dog age* remained as the variable of most importance, but other variables of lesser importance were identified that had not been seen on the original models (including *breed size category*, *owner age*, *education level* and *urban setting*), whilst a limited effect of diet remained (i.e., the *vegan diet* variable was only the 12th most important predictor variable). The significance of any associations between these variables owner perceptions of health are not known. Nevertheless, since predictive accuracy of all models (both multiple logistic regression and XGBoost) utilising just the *owner-animal metadata* was poor, these analyses should be considered exploratory and any conclusions drawn should be made with caution.

A final approach we took, to ensure we had fully accounted for any possible effect of the *dog diet vegan* variable, was to add it back into the best-fit models for both the *any health problem* and *serious illness* outcome variables. In both cases, this additional predictor variable did not

improve the performance of either model and, interestingly, models with equivalent performance could be created when the *owner diet vegan* was instead added.

The main limitations to the current study have already been discussed above, including the fact that owner-reported health was the main outcome measure and the fact that the study population was not representative of the general dog-owning population. Further, significant data pre-processing was required to ensure adequate group sizes for statistical analyses. The disadvantage of such an approach was that many smaller groups needed to be excluded, meaning that we might have missed some variables with a potential impact on owner-reported canine health. That said, in preliminary testing, we coded several of our variables in different ways (e.g., *dog age* as a continuous or ordinal 5-category variable; *veterinary visits* as a 4- or 5-category ordinal variable), and selected the approach that produced the best model fit. Therefore, whilst some genuine effects from small categories might still have been missed, the final best-fit models we selected were those that most closely fitted the data.

A third limitation was the fact that we did not contact the authors of the original study before conducting our data analyses, to clarify any uncertainties with the dataset and seek guidance on our planned approach. We chose not to do so because the original paper was well written, with clearly presented study methodology and results. Further, since independent replication is critical to the scientific method, it was arguably better not to have direct contact with the primary study authors when conducting statistical analyses.

A fourth limitation was that the outcome variables tested in the current study were derived from only one of the health metrics examined in the original study, namely owner-reported dog health [15]. Other health metrics reported by Knight *et al.* included asking owners "to report what they believed their veterinarian's assessment [of their dog's health] to be" [15] and owner reports of 22 possible health disorders, grouped into broad categories (e.g., allergy, cancer/tumours, hormonal, skin/coat) [15]. Our conclusions might have differed if one or more of these alternative health metrics had been analysed, but we decided against this because, in our opinion, all were inferior measures. Not only were these metrics also based on subjective owner reports, but owners were required to speculate on what their veterinarian's opinion about the health of their dog would be. Further, there were no clear definitions of the health disorder categories, meaning that these would likely be broad, comprising many different diseases with differing aetiology and severity. Finally, very few (≤50) dogs were reported to be suffering from many (12/22) of the health-disorders studied [15] and, therefore, any statistical analyses would likely be underpowered to identify associations with predictor variables. Whilst further work could be considered, where these additional metrics are analysed using the techniques of the current study, it is questionable as to whether anything further would be gained. Instead, it would be preferable to conduct a prospective study utilising objective, veterinary-derived measures of health, as discussed above.

A final limitation was the fact that there might be confounding between healthcare variables (e.g., *veterinary visits* and *received medication*) and dietary choices. For example, owners who are sceptical about conventional diets and instead choose to feed either vegan or raw food, might also be less willing either visiting their veterinarian or allowing their dog to receive medication. In support of this, negative associations were identified in the current study between feeding a raw diet and both veterinary visit frequency and the use of medications; further, in the previous study, dogs fed raw diets were less likely to be neutered than dogs fed conventional diets [15]. However, results were inconsistent for dogs fed vegan diets; whilst there was a negative association between feeding a vegan diet and the odds of a dog receiving medications, there was no association with frequency of veterinary visits. Further, in the previous study, dogs fed vegan diets were more likely to be neutered than those fed conventional diets [15].

Given these complexities, further research would be required to investigate associations between different diet choices and healthcare variables.

## Conclusions

Variables most strongly associated with owner perceptions of health are dog age and health-care variables including number of veterinary visits, receiving medication and being switched to a therapeutic diet. These results extend the findings of previously-published research [15] and suggest that, when other variables are accounted for, the association between dog diet choice and owner-perceived canine health is negligible.

## Supporting information

**S1 Fig. Owner-dog metadata visualised by uniform manifold approximation and projection with density-based spatial clustering.** Owner-pet metadata were pre-processed with all factors as numeric and subject to dimension reduction with the UMAP projection technique. Owner variables included in this visualisation were *diet*, *sex*, *location*, *education* and *income*; animal variables included were *age*, *sex*, *neuter status*, *diet* and *breed size category*. Healthcare variables were not included. Each individual row of the data (owner-dog combination) contributes one UMAP x and y coordinate, analogous to PC1 and PC2 in a principal component analysis. Four or five distinct clusters were evident. Points are colour-coded by *owner gender* (a), *income* (b), *location* (c), *setting* (d), *education* € and *decision-maker status* (f) as indicated in the legend.
(PDF)

**S2 Fig. Owner-dog metadata visualised by uniform manifold approximation and projection with density-based spatial clustering.** Owner-pet metadata were pre-processed with all factors as numeric and subject to dimension reduction with the UMAP projection technique. Owner variables included in this visualisation were *diet*, *sex*, *location*, *education* and *income*; animal variables included were *age*, *sex*, *neuter status*, *diet* and *breed size category*. Healthcare variables were not included. Each individual row of the data (owner-dog combination) contributes one UMAP x and y coordinate. Four or five distinct clusters were evident. Points are colour-coded by *owner age* (a), *animal career* (b), *dog age* (c), *breed size category* (d), *dog sex* € and *neuter status* (f) as indicated in the legend.
(PDF)

**S3 Fig.** Multiple binary logistic regression model, on data from all owners, with *any health problem* as the outcome variable and including either all owner, animal and including either all healthcare variables (a) or only the owner-animal metadata (b). The dots represent the odds ratio for each variable, whilst the bars represent 99% confidence intervals (99%-CI). Variables where the 99%-CI range does not include 1.0 (vertical dotted line) are depicted in red, whilst those that include 1.0 are depicted in blue. Note the logarithmic scale for the X-axis.
(TIF)

**S4 Fig.** Multiple binary logistic regression model with *significant illness* as the outcome variable, on data from owners who were primary carers, either all owner, animal and including either all healthcare variables (a) or only the owner-animal metadata (b). The dots represent the odds ratio for each variable, whilst the bars represent 99% confidence intervals (99%-CI). Variables where the 99%-CI range does not include 1.0 (vertical dotted line) are depicted in red, whilst those that include 1.0 are depicted in blue. Note the logarithmic scale for the X-axis.
(TIF)

**S5 Fig. All-variable XGBoost model on the *any health problem* binary using the all-owner dataset.** (a) Receiver operating characteristic curve of a prediction model containing all variables (owner, animal and healthcare). This shows the increasing true positive and false positive rates, with decrease of the threshold probability for prediction of *any health problem*. Prediction accuracy was good as assessed by ROC analysis (area under curve 0.838, 99%-CI: 0.797–0.879). Acceptance threshold is indicated by the colour bar on the right-hand side and shown at discrete points on the curve. (b) Graph depicting the relative contribution of predictor variables to the all-variable XGBoost model for the *any health problem* binary. 'Importance' represents fractional contribution of each feature to the model, based on the total gain from including each feature; 'cover' represents the number of observations related to this feature in the model); 'frequency' represents the relative number of times a feature has been used in trees). Variables are organised in order of importance left to right, based on the sum of the 3 metrics, whilst variables that are not shown did not contribute to the final model.
(TIF)

**S6 Fig. Reduced (owner-animal metadata) XGBoost model on the *any health problem* binary using the all-owner dataset.** (a) Receiver operating characteristic curve of a reduced prediction model only containing owner and animal variables. This shows the increasing true positive and false positive rates, with decrease of the threshold probability for prediction of *any health problem*. Prediction accuracy was moderate, as assessed by ROC analysis (AUC 0.682, 99%-CI: 0.628–0.737). Acceptance threshold is indicated by the colour bar on the right-hand side and shown at discrete points on the curve. The fact that the prediction thresholds are all low ($< 0.5$) shows that this model struggles to predict health issues. (b) Graph depicting the relative contribution of predictor variables to the all-variable XGBoost model for the *any health problem* binary. 'Importance' represents fractional contribution of each feature to the model, based on the total gain from including each feature; 'cover' represents the number of observations related to this feature in the model); 'frequency' represents the relative number of times a feature has been used in trees). Variables are organised in order of importance left to right, based on the sum of the 3 metrics, whilst variables that are not shown did not contribute to the final model.
(TIF)

**S7 Fig. All-variable XGBoost model on the *significant illness* binary using the all-owner dataset.** (a) Receiver operating characteristic curve of a prediction model containing all variables (owner, animal and healthcare). This shows the increasing true positive and false positive rates, with decrease of the threshold probability for prediction of *significant illness*. Prediction accuracy was good, as assessed by ROC analysis (area under curve 0.884, 99%-CI: 0.796–0.972). Acceptance threshold is indicated by the colour bar on the right-hand side and shown at discrete points on the curve. (b) Graph depicting the relative contribution of predictor variables to the all-variable XGBoost model for the *significant illness* binary. 'Importance' represents fractional contribution of each feature to the model, based on the total gain from including each feature; 'cover' represents the number of observations related to this feature in the model); 'frequency' represents the relative number of times a feature has been used in trees). Variables are organised in order of importance left to right, based on the sum of the 3 metrics, whilst variables that are not shown did not contribute to the final model.
(TIF)

**S8 Fig. Reduced (owner-animal metadata) XGBoost model on the *significant illness* binary using the all-owner dataset.** (a) Receiver operating characteristic curve of a reduced prediction model only containing owner and animal variables. This shows the increasing true

positive and false positive rates, with decrease of the threshold probability for prediction of *significant illness*. Prediction accuracy was moderate, as assessed by ROC analysis (area under curve 0.664, 99%-CI: 0.535–0.693). Acceptance threshold is indicated by the colour bar on the right-hand side, and also shown at discrete points on the curve. (b) Graph depicting the relative contribution of predictor variables to the all-variable XGBoost model for the *significant illness* binary. 'Importance' represents fractional contribution of each feature to the model, based on the total gain from including each feature; 'cover' represents the number of observations related to this feature in the model); 'frequency' represents the relative number of times a feature has been used in trees). Variables are organised in order of importance left to right, based on the sum of the 3 metrics, whilst variables that are not shown did not contribute to the final model.
(TIF)

**S1 Table. Results of simple (i.e., univariable) binary logistic regression analyses examining associations between owner, animal and veterinary variables and the *any health problem binary* for all owners.**
(DOCX)

**S2 Table. Results of simple (i.e., univariable) binary logistic regression analyses examining associations between owner, animal and veterinary variables and the *significant illness* binary for all owners.**
(DOCX)

**S3 Table. Kendall's tau correlation coefficients for the naïve associations amongst owner-animal metadata variables in Fig 2.**
(NUMBERS)

**S4 Table. Respective log *P*-values for the Kendall's tau correlations amongst owner-animal metadata variables in Fig 2 and S1 Table.**
(NUMBERS)

**S5 Table. Final multiple binary logistic regression model examining associations between owner, animal and veterinary variables and the presence of *any health problem* as reported by the owner.**
(DOCX)

**S6 Table. Final multiple binary logistic regression model examining associations between owner, animal and veterinary variables and the presence of a significant or serious illness as reported by the owner.**
(DOCX)

**S1 File. Statistical report containing details of all data pre-processing steps to create the dataset for primary decision-makers, as well as initial data visualisation using UMAP.**
(HTML)

**S2 File. Statistical report containing details of all data pre-processing steps to create the dataset for all owners.**
(HTML)

**S3 File. Statistical report containing details of all statistical analyses for the Kendall's rank correlation, and also XGBoost modelling for the *any health problem* outcome using data from primary decision makers.**
(HTML)

**S4 File. Statistical report containing details of all statistical analyses for the Kendall's rank correlation, and also XGBoost modelling for the *any health problem* outcome using using data from all owners.**
(HTML)

**S5 File. Statistical report containing details of XGBoost modelling for the *significant illness* outcome using data from primary decision makers.**
(HTML)

**S6 File. Statistical report containing details of XGBoost modelling for the *significant illness* outcome using data from all owners.**
(HTML)

**S7 File. Statistical report containing details of binary logistic regression modelling for the *any health problem* variable using data from primary decision makers.**
(HTML)

**S8 File. Statistical report containing details of binary logistic regression modelling for the *any health problem* outcome using data from all owners.**
(HTML)

**S9 File. Statistical report containing details of binary logistic regression modelling for the *significant illness* variable using data from primary decision makers.**
(HTML)

**S10 File. Statistical report containing details of binary logistic regression modelling for the *significant illness* variable using data from all owners.**
(HTML)

## Acknowledgments

The authors wish to acknowledge the authors of the original paper [15] for providing making their data freely available to other researchers for further analysis.

## Author Contributions

**Conceptualization:** Richard Barrett-Jolley, Alexander J. German.

**Data curation:** Richard Barrett-Jolley, Alexander J. German.

**Formal analysis:** Richard Barrett-Jolley, Alexander J. German.

**Methodology:** Richard Barrett-Jolley, Alexander J. German.

**Project administration:** Richard Barrett-Jolley, Alexander J. German.

**Software:** Richard Barrett-Jolley, Alexander J. German.

**Validation:** Richard Barrett-Jolley, Alexander J. German.

**Visualization:** Richard Barrett-Jolley, Alexander J. German.

**Writing – original draft:** Richard Barrett-Jolley, Alexander J. German.

**Writing – review & editing:** Richard Barrett-Jolley, Alexander J. German.

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
