## [Decision Letter · Decision Letter 0]

6 Jun 2023

PONE-D-22-34984Variables associated with owner perceptions of the health of their dog: further analysis of data from a large international surveyPLOS ONE

Dear Dr. German,

Thank you for submitting your manuscript to PLOS ONE. After careful consideration, we feel that it has merit but does not fully meet PLOS ONE’s publication criteria as it currently stands. Therefore, we invite you to submit a revised version of the manuscript that addresses the points raised during the review process.

The two reviewers come up with conflicting decisions; reviewer#1 accept and reviewer #2 reject. Reviewer #2 despite  has an obvious conflict of interest, has made detailed comments BUT methodology and statistical analysis needs a point by point critical response. Obviously this article needs to address the issues raised by reviewers #2 before reaching a final decision or invite additional reviewers. 

We look forward to receiving your revised manuscript.

Kind regards,

Gizat Almaw

Academic Editor

PLOS ONE

Journal Requirements:

"AJG is an employee of the University of Liverpool, but his position is financially-supported by Royal Canin.  AJG has also received financial remuneration and gifts for providing educational material, speaking at conferences, and consultancy work, all unrelated to the current study.  Royal Canin had no involvement in any aspect of the current work including study design, data analysis, drafting the manuscript or the decision to submit the work for publication.

RBJ is an employee of the University of Liverpool, whose salary is financially-supported by the Higher Education Funding Council for England.  RBJ also receives a stipend from United Kingdom Research and Innovation for work as a panellist for the Biotechnology and Biological Sciences Research Council."

Reviewers' comments:

Reviewer's Responses to Questions

**Comments to the Author**

1. Is the manuscript technically sound, and do the data support the conclusions?

Reviewer #1: Yes

Reviewer #2: No

2. Has the statistical analysis been performed appropriately and rigorously? 

Reviewer #1: Yes

Reviewer #2: No

3. Have the authors made all data underlying the findings in their manuscript fully available?

Reviewer #1: Yes

Reviewer #2: Yes

4. Is the manuscript presented in an intelligible fashion and written in standard English?

Reviewer #1: Yes

Reviewer #2: Yes

5. Review Comments to the Author

Reviewer #1: Happy to have reviewed the paper titled "variables associated with owner perceptions of the health of their dog: further analysis of data from a large international survey".

A timely and well-written paper with extensive analysis of data.

The authors have produced additional information on owner-reported dog health status and their diet using different statistical techniques to create models that best predicted owner opinions of health, and to identify the relative importance of the variables that contributed to the final model.

Reviewer #2: This authors of this current study responded to our prior Knight et al (2022) study (https://doi.org/10.1371/journal.pone.0265662), after reanalysing the dataset we collected, which is available open access.

For my review please see the attachment.

6. PLOS authors have the option to publish the peer review history of their article (what does this mean?). If published, this will include your full peer review and any attached files.

Reviewer #1: No

Reviewer #2: **Yes: **Andrew Knight

---

## [Author Response · Author response to Decision Letter 0]

14 Jun 2023

We are extremely grateful to both reviewers for their comments. Please see the rebuttal document ("Response to Reviewers"), which details our responses and how we have revised the manuscript to take them into account.

---

## [Decision Letter · Decision Letter 1]

3 Nov 2023

PONE-D-22-34984R1Variables associated with owner perceptions of the health of their dog: further analysis of data from a large international surveyPLOS ONE

Dear Dr. German,

Thank you for submitting your manuscript to PLOS ONE. After careful consideration, we feel that it has merit but does not fully meet PLOS ONE’s publication criteria as it currently stands. Therefore, we invite you to submit a revised version of the manuscript that addresses the points raised during the review process.

We look forward to receiving your revised manuscript.

Kind regards,

Juan J Loor

Academic Editor

PLOS ONE

Reviewers' comments:

Reviewer's Responses to Questions

**Comments to the Author**

1. If the authors have adequately addressed your comments raised in a previous round of review and you feel that this manuscript is now acceptable for publication, you may indicate that here to bypass the “Comments to the Author” section, enter your conflict of interest statement in the “Confidential to Editor” section, and submit your "Accept" recommendation.

Reviewer #1: All comments have been addressed

Reviewer #2: (No Response)

Reviewer #3: (No Response)

2. Is the manuscript technically sound, and do the data support the conclusions?

Reviewer #1: Yes

Reviewer #2: Partly

Reviewer #3: Partly

3. Has the statistical analysis been performed appropriately and rigorously? 

Reviewer #1: Yes

Reviewer #2: No

Reviewer #3: No

4. Have the authors made all data underlying the findings in their manuscript fully available?

Reviewer #1: Yes

Reviewer #2: Yes

Reviewer #3: Yes

5. Is the manuscript presented in an intelligible fashion and written in standard English?

Reviewer #1: Yes

Reviewer #2: Yes

Reviewer #3: Yes

6. Review Comments to the Author

Reviewer #1: None

Reviewer #2: The authors of this current study responded to our prior Knight et al (2022) study (https://doi.org/10.1371/journal.pone.0265662), after reanalysing the dataset we collected, which is available open access. Please see attachment for my review.

Reviewer #3: The authors present a secondary analysis of a dataset from a previously published study on dog diet and health outcomes. The authors' goal was to incorporate more dog and owner information into models to account for possible confounding. The authors provide extensive detail in supplemental material (including statistical code), in addition to the manuscript itself, enabling the readers to follow along and understand exactly what was done. Although the authors approached the question using multiple approaches, generally arriving at similar conclusions, I do have some concerns about the approach. The manuscript will be strengthened if the authors consider the following points:

1. The authors use UMAP and DBSCAN as exploratory tools to assess the underlying structure of the data. They state they coerced all variables to be numeric in order to use these techniques. It is not clear to me that this is a sufficient way to handle the non-ordinal categorical variables (such as diet, gender and location). All examples I looked up for UMAP and DBSCAN focused only on variables that were inherently numeric and since these methods appear to have some sort of "distance" that is calculated, the numeric values matter. Do authors arrive at the same clusters if other orderings (numeric assignments) of the non-ordinal data are used?

2. Using healthcare variables (# of vet visits, prescribed medication by vet, on therapeutic diet) as predictors of owner perceived health of their pet seems not all that informative, since likely these variables are a part of an owner's assessment of the health of their pet. For example, I find it hard to believe that very many owners who had to take their pet to the vet numerous times in a year would characterize their pet as healthy. So, it seems obvious that these would end up being the most predictive of the outcome. Since, to me, they seem to be part of the outcome, why consider them as predictors at all? Authors could instead use those as alternative outcome measures of health and see how diet and dog and owner characteristics are associated with perceived health.

Minor points:

1. line 43-44: "albeit having limited although... was limited" is awkwardly phrased.

2. line 50: "that diet the fed" is awkwardly phrased

3. line 318 - it appears as though authors have switched vegan and vegetarian (Table 1 has 493 vegetarians and 221 vegans, while the text has them reversed).

4. lines 325-328: the numbers given for owners identifying as vegan (324) does not make sense given the numbers in Table 1 - even if the labels are reversed in Table 1, 324 would not be 56% of all vegan owners. Also, the numbers of vegetarian and vegan diets for the dogs do not match those presented in Table 1. Finally, it is not clear why the number of observations in the cluster mentioned in this section changes depending on whether it is being illustrated by owner diet or by dog diet (381 vs 383).

5. In supplemental figures S1 and S2, why are authors not using the processed variables (with collapsed categories) for client age and education as described in lines 161-163?

6. How were diets for those on a therapeutic diet handled? Those would be at the recommendation of the veterinarian, so not exactly the complete choice of the owner.

7. To avoid confusion in the usage of the variables that are derived from the same information, authors should only mention the "selected" one for consideration in the multivariable models. For example, in lines 395-396, authors do not need to mention both breed size category and giant breed - their methods describe how they chose the "best" variable. I do wonder, however, in a situation like this where giant breed results in a lower BIC, if authors are in fact combining too many breeds together by just using this single binary variable to represent breed.

8. The list of variables considered for the multiple regression in lines 435-437 does not seem to correspond to the results in Table 3 and the approach described in the Methods section for selection of those variables. For example, owner gender, animal related career and neuter status all seem to be variables that should be considered...and it is not clear why giant breed is being considered. The same can be said for the list of variables give in lines 513-515. Authors need to make sure what they state matches their results and the approach they say they took.

9. captions to figures 5, 6, 8, 9 - authors refer to a false-negative rate (lines 488, 501,560, 573), but these should be false positive rates based on your figure.

10. lines 603: authors state that dog diet was not associated with the significant illness outcome, but based on Table 4, raw diet was.

7. PLOS authors have the option to publish the peer review history of their article (what does this mean?). If published, this will include your full peer review and any attached files.

Reviewer #1: No

Reviewer #2: **Yes: **Andrew Knight

Reviewer #3: No

---

## [Author Response · Author response to Decision Letter 1]

7 Dec 2023

We have responded in detail to all reviewer comments in the point-by-point response document.

---

## [Decision Letter · Decision Letter 2]

11 Mar 2024

PONE-D-22-34984R2Variables associated with owner perceptions of the health of their dog: further analysis of data from a large international surveyPLOS ONE

Dear Dr. German,

Thank you for submitting your manuscript to PLOS ONE. After careful consideration, we feel that it has merit but does not fully meet PLOS ONE’s publication criteria as it currently stands. Therefore, we invite you to submit a revised version of the manuscript that addresses the points raised during the review process.

We look forward to receiving your revised manuscript.

Kind regards,

Juan J Loor

Academic Editor

PLOS ONE

Journal Requirements:

Reviewers' comments:

Reviewer's Responses to Questions

**Comments to the Author**

1. If the authors have adequately addressed your comments raised in a previous round of review and you feel that this manuscript is now acceptable for publication, you may indicate that here to bypass the “Comments to the Author” section, enter your conflict of interest statement in the “Confidential to Editor” section, and submit your "Accept" recommendation.

Reviewer #2: (No Response)

2. Is the manuscript technically sound, and do the data support the conclusions?

Reviewer #2: Partly

3. Has the statistical analysis been performed appropriately and rigorously? 

Reviewer #2: Yes

4. Have the authors made all data underlying the findings in their manuscript fully available?

Reviewer #2: Yes

5. Is the manuscript presented in an intelligible fashion and written in standard English?

Reviewer #2: Yes

6. Review Comments to the Author

Reviewer #2: I appreciate the many improvements made within the current version.

Minor revisions are required - please see attached.

7. PLOS authors have the option to publish the peer review history of their article (what does this mean?). If published, this will include your full peer review and any attached files.

Reviewer #2: **Yes: **Andrew Knight

---

## [Author Response · Author response to Decision Letter 2]

13 Mar 2024

Our responses to the editorial and reviewer comments can be found in the "Response to Reviewers" document

---

## [Editor Report · Decision Letter 3]

3 Apr 2024

Variables associated with owner perceptions of the health of their dog: further analysis of data from a large international survey

PONE-D-22-34984R3

Dear Dr. German,

We’re pleased to inform you that your manuscript has been judged scientifically suitable for publication and will be formally accepted for publication once it meets all outstanding technical requirements.

Kind regards,

Juan J Loor

Academic Editor

PLOS ONE

Additional Editor Comments (optional):

The paper has undergone a fair amount of revisions and I believe the authors have addressed all outstanding issues.